# Policy Optimization with Distributional Constraints: An Optimal Transport View

## Abstract

We consider constrained policy optimization in Reinforcement Learning, where the constraints are in form of marginals on state visitations and global action executions. Given these distributions, we formulate policy optimization as unbalanced optimal transport over the space of occupancy measures. We propose a general purpose RL objective based on Bregman divergence and optimize it using Dykstra's algorithm when the transition model is known. The approach admits an actor-critic algorithm for when the state or action space is large, and only samples from the marginals are available. We discuss applications of our approach and provide demonstrations to show the effectiveness of our algorithm.

## 1 Introduction

In reinforcement learning (RL), policy optimization seeks an optimal decision making strategy, known as a policy (Bertsekas, 1995; Williams, 1992; Sutton et al., 2000). Policies are typically optimized in terms of accumulated rewards with or without constraints on actions and/or states associated with an environment (Altman, 1999).

Policy optimization has many challenges; perhaps the most basic is the constraint on flow of state-action visitations called *occupancy measures*. Indeed, formulating RL as a linear programming problem, occupancy measures appear as an explicit constraint on the optimal policy (Puterman, 2014). The constraint-based formulation suggests the possibility of implementing a broader set of objectives and constraints, such as entropy regularization (Peters et al., 2010; Neu et al., 2017; Nachum & Dai, 2020) and cost-constrained MDPs (Altman, 1999).

Considering the reward function as negative cost of assigning an action to a state, we view RL as a stochastic assignment problem. We formulate policy optimization as an unbalanced optimal transport on the set of occupancy measures. Where Optimal Transport (OT) (Villani, 2008) is the problem of adapting two distributions on possibly different spaces via a cost function, unbalanced OT relaxes the marginal constraints on OT to arbitrary measures through penalty functions (Liero et al., 2017; Chizat et al., 2017).

We therefore define distributionally-constrained reinforcement learning as a problem of optimal transport. Given baseline marginals over states and actions, policy optimization is unbalanced optimal transport adapting the state marginal to action marginal via the reward function. Built upon mathematical tools of OT, we generalize the RL objective to the summation of a Bregman divergence $D_\Gamma$ and any number of arbitrary lower-semicontinuous convex functions $\phi_i'$s and an initial distribution $\xi$ as

$$\min_{\mu \in \Delta} D_\Gamma(\mu|\xi) + \sum_i^N \phi_i(\mu),$$

where $\Delta$ is the set of occupancy measures (joint distributions on states and actions generated from policies). We optimize this objective with *Dykstra's algorithm* (Dykstra, 1983) which is a method of iterative projections onto general closed convex constraint sets. Under Fenchel duality, this algorithm allows decomposition of the objective into Bregman projections on the subsets corresponding to each function.

As particular case, we can regularize over the state space distribution and/or the global action execution distribution of the desired occupancy measures. Given the reward function $r$ and divergence functions $D_{\psi_1}$ and $D_{\psi_2}$, our formulation allows constraints on the policy optimization problem in terms of distributions on state visitations $\rho'$ and/or action executions $\eta'$ as:

$$\max_{\mu \in \Delta} \mathbb{E}_\mu[r] - \epsilon_1 D_{\psi_1}\left(\mu \mathbf{1} \mid \rho'\right) - \epsilon_2 D_{\psi_2}\left(\mu^T \mathbf{1} \mid \eta'\right).$$

Marginal constraints of $\rho'$ and $\eta'$ can arise in various settings in policy optimization: problem constraints of budget on actions or safety constraints on state visitations (Altman, 1999; Zhang et al., 2020a), empirical state and action distributions from an expert demonstrations or for regularization purposes in an iterative policy optimization algorithm (Neu et al., 2017)

We also propose an actor-critic algorithm with function approximation for large scale RL, for when we have access to samples from a baseline policy (off-policy sampling or imitation learning) and samples from the constraint marginals.

The structure of the paper is as follows: In the Section 2 we briefly present the preliminaries on (unbalanced) optimal transport and policy optimization in reinforcement learning. In Section 3 we introduce a general objective with Bregman divergence for policy optimization and provide Dykstra iterations as a general primal algorithm for optimizing this objective. Section 4 discusses distributionally constrained policy optimization with unbalanced OT and its applications. In this section, we also provide an actor critic algorithm for large scale RL. We conclude with demonstrations of the distributional constraints in Section 5 and discussion on related works in Section 6.

## 2 Notation and Preliminaries

For any finite set $\mathcal{X}$, let $\mathcal{M}(\mathcal{X})$ be the set of probability distributions on $\mathcal{X}$. We denote the indicator function on set $\mathcal{X}$ by $\delta_{\mathcal{X}}(x) = \begin{cases} 0 & \text{if } x \in \mathcal{X} \\ \infty & \text{otherwise} \end{cases}$. For $p \in \mathcal{M}(\mathcal{X})$, we define the *entropy* map $\mathcal{H}(p) = \mathbb{E}_{x \sim p}[\log p(x) - 1]$, and denote the *Kullback-Leibler (KL) divergence* between two positive functions $p, q$ by $\text{KL}(p|q) = \sum_{x \in \mathcal{X}} p(x) \log\left(\frac{p(x)}{q(x)}\right) - p(x) + q(x)$. If $p, q \in \mathcal{M}(\mathcal{X})$, for a given convex function $\psi : \mathcal{X} \to \mathbb{R}$ with $\psi(1) = 0$, we define $\psi$-divergence: $D_\psi(p|q) = \sum_x \psi\left(\frac{p(x)}{q(x)}\right) q(x)$. In particular, for $\psi(x) = x \log(x)$, $D_\psi(p|q) = \text{KL}(p|q)$. We also use $\langle \cdot, \cdot \rangle$ as the natural inner product on $\mathcal{X}$. Through out the paper by $\mathbf{1}_{\mathcal{X}}$ we denote a vector with elements one over set $\mathcal{X}$ or just $\mathbf{1}$ if the context is clear.

### 2.1 Optimal Transport

Given measures $a \in \mathcal{M}(\mathcal{X})$, $b \in \mathcal{M}(\mathcal{Y})$ on two sets $\mathcal{X}$ and $\mathcal{Y}$, with a cost function $C : \mathcal{X} \times \mathcal{Y} \to \mathbb{R}$, Kantorovich *Optimal Transportation* is the problem of finding stochastic optimal plan $\mu \in \mathcal{M}(\mathcal{X} \times \mathcal{Y})$:

$$\min_\mu \mathbb{E}_\mu[C(x, y)] + \delta_{\{a\}}\left(\mu \mathbf{1}_{\mathcal{Y}}\right) + \delta_{\{b\}}\left(\mu^T \mathbf{1}_{\mathcal{X}}\right).$$

When $\mathcal{X} = \mathcal{Y}$ and $C$ is derived from a metric on $\mathcal{X}$, this optimization defines a distance function on measure space $\mathcal{M}(\mathcal{X})$, called Wasserstein distance (Villani, 2008).

*Unbalanced Optimal Transport* replaces hard constraints $\delta_{\{a\}}$ and $\delta_{\{b\}}$, with penalty functions

$$\min_\mu \mathbb{E}_\mu[C(x, y)] + \epsilon_1 D_{\psi_1}(\mu \mathbf{1}|a) + \epsilon_2 D_{\psi_2}(\mu^T \mathbf{1}|b),$$

where $\epsilon_1, \epsilon_2$ are positive scalars. This formulation also extends OT to measures of arbitrary mass. As $\epsilon_1, \epsilon_2 \to \infty$, the unbalanced OT approaches Kantorovich OT problem (Liero et al., 2017; Chizat et al., 2017).

To speed up the daunting computational costs of standard algorithms, an entropy term $\mathcal{H}(\mu)$ is usually added to the (U)OT objective to apply scaling algorithms (Cuturi, 2013; Chizat et al., 2017). When $\mathcal{X}$ and

$\mathcal{Y}$ are large or continuous spaces, we usually have access to samples from $a, b$ instead of the actual measures. Stochastic approaches usually add a relative entropy $\mathrm{KL}(\mu \mid a \otimes b)$, instead of $\mathcal{H}(\mu)$ in order to take advantage of the Fenchel dual of the (U)OT optimization and estimate the objective from samples out of $a, b$ (Aude et al., 2016; Seguy et al., 2018).

## 2.2 Reinforcement Learning

Consider a discounted MDP $(\mathcal{S}, \mathcal{A}, P, r, \gamma, p_0)$, with finite state space $\mathcal{S}$, finite action space $\mathcal{A}$, transition model $P : \mathcal{S} \times \mathcal{A} \to \mathcal{M}(\mathcal{S})$, initial distribution $p_0 \in \mathcal{M}(\mathcal{S})$, deterministic reward function $r : \mathcal{S} \times \mathcal{A} \to \mathbb{R}$ and discount factor $\gamma \in [0, 1)$. Letting $\Pi$ be the set of stationary policies on the MDP, for any policy $\pi : \mathcal{S} \to \mathcal{M}(\mathcal{A})$, we denote $P^\pi : \mathcal{S} \to \mathcal{M}(\mathcal{S})$ to be the induced Markov chain on $S$. In policy optimization, the objective is

$$\max_{\pi \in \Pi} \sum_{s,a} \rho^\pi(s) \pi(a|s) r(s, a), \tag{1}$$

where $\rho^\pi(s) = (1 - \gamma) \sum_{t=0}^\infty \gamma^t \Pr(s_t = s | \pi)$ is the discounted stationary distribution of $P^\pi$. For a policy $\pi$, we define its *occupancy measure* $\mu^\pi \in \mathcal{M}(\mathcal{S} \times \mathcal{A})$, as $\mu^\pi(s, a) = \rho^\pi(s)\pi(a|s)$. Let $\Delta := \{\mu^\pi : \pi \in \Pi\}$ be the set of occupancy measures of $\Pi$, the following lemma bridges the two sets $\Pi$ and $\Delta$:

**Lemma 2.1.** (Syed et al., 2008)[Theorem 2, Lemma2]

(i) $\Delta = \{\mu \in \mathcal{M}(\mathcal{S} \times \mathcal{A}) : \sum_a \mu(s, a) = (1 - \gamma)p_0(s) + \gamma \sum_{s',a'} P(s|s', a')\mu(s', a') \ \forall s\}$

(ii) $\pi^\mu(a|s) := \mu(s, a) / \sum_a \mu(s, a)$ is a bijection from $\Delta$ to $\Pi$.

By multiplying $\pi = \pi^\mu$ to both sides of the equation in (i), one can obtain $\Delta = \{\mu : \mu \geq 0, A^\mu \mu = b^\mu\}$ where $A^\mu = \mathbb{I} - \gamma P_\mu{}^T$ and $b^\mu = (1 - \gamma)\pi p_0$. In the rest of paper, we may drop the superscripts $\mu$ and $\pi$, when the context is clear. Rewriting the policy optimization objective equation 1 in terms of $\mu$, we get

$$\max_{\mu \in \Delta} \mathbb{E}_\mu[r] = \max_\mu \mathbb{E}_\mu[r] - \delta_{\{b^\mu\}}(A^\mu \mu). \tag{2}$$

The entropy-regularized version of objective equation 2, relative to a given baseline $\mu' \in \Delta$, is also studied (Peters et al., 2010; Neu et al., 2017):

$$\max_{\mu \in \Delta} \mathbb{E}_\mu[r] - \epsilon \mathrm{KL}(\mu|\mu') \equiv \min_{\mu \in \Delta} \mathrm{KL}\left(\mu|\mu' e^{r/\epsilon}\right), \tag{3}$$

where $\epsilon$ is the regularization coefficient.

By lemma 2.1, one can decompose the regularization term in equation 3 as

$$\mathrm{KL}(\mu|\mu') = \mathrm{KL}\left(\sum_a \mu \Big| \sum_a \mu'\right) + \mathbb{E}_{\rho^\mu}\left[\mathrm{KL}\left(\pi|\pi'\right)\right], \tag{4}$$

with the first term penalizing the shift on state distributions and the second penalty is over average shift of policies for every state. Since the goal is to optimize for the best policy, one might consider only regularizing relative to $\pi'$ as in (Schulman et al., 2015; Neu et al., 2017)

$$\max_{\mu \in \Delta} \mathbb{E}_\mu[r] - \epsilon \mathbb{E}_{\rho^\mu}\left[\mathrm{KL}(\pi|\pi')\right]. \tag{5}$$

One can also regularize objective equation 2 by $\mathcal{H}(\mu)$ as

$$\max_{\mu \in \Delta} \mathbb{E}_\mu[r] - \epsilon \mathcal{H}(\mu) \equiv \min_{\mu \in \Delta} \mathrm{KL}(\mu|e^{r/\epsilon}), \tag{6}$$

which encourages exploration and avoids premature convergence (Haarnoja et al., 2017; Schulman et al., 2018; Ahmed et al., 2019).

# 3 A General RL Objective with Bregman Divergence

In this section, we propose a general RL objective based on Bregman divergence and optimize using Dykstra's algorithm when the transition model is known. Let $\Gamma$ be a strictly convex, smooth function on relint(dom($\Gamma$)), the relative interior of its domain, with convex conjugate $\Gamma^*$. For any $(\mu, \xi) \in \text{dom}(\Gamma) \times \text{int}(\text{dom}(\Gamma))$, we define the Bregman divergence

$$D_\Gamma(\mu|\xi) = \Gamma(\mu) - \Gamma(\xi) - \langle \nabla\Gamma(\xi), \mu - \xi \rangle.$$

Given $\xi$, we consider the optimization

$$\min_\mu D_\Gamma(\mu|\xi) + \sum_i^N \phi_i(\mu), \tag{7}$$

where $\phi_i'$s are proper, lower-semicontinuous convex functions satisfying

$$\cap_i^N \text{relint}(\text{dom}(\phi_i)) \cap \text{relint}(\text{dom}(\Gamma)) \neq \emptyset. \tag{8}$$

Let dom($\Gamma$) be the simplex on $\mathcal{S} \times \mathcal{A}$, regularized RL algorithms in Section 2.2 can be observed as instances of optimization equation 7:

• Given a baseline $\mu'$, setting $\Gamma = \mathcal{H}$, $\phi_1(\mu) = \delta_{\{b^\mu\}}(A^\mu\mu)$, $\xi = \mu'e^{r/\epsilon}$, recovers objective equation 3.

• Similarly, as discussed in (Neu et al., 2017), $\Gamma(\mu) = \sum_{s,a} \mu(s,a) \log \mu(s,a)/\sum_a \mu(s,a)$, $\phi_1 = \mathbb{E}_\mu[r/\epsilon]$, $\phi_2(\mu) = \delta_{\{b^\mu\}}(A^\mu\mu)$, and $\xi = \mu'$ recovers obj. equation 5.

• Further, $\Gamma = \mathcal{H}$, $\phi_1(\mu) = \delta_{\{b^\mu\}}(A^\mu\mu)$, and $\xi = e^{r/\epsilon}$, entropy-regularizes the occupancy measure in objective equation 6.

The motivation behind using Bregman divergence is to generalize the KL divergence regularization usually used in RL algorithms. Moreover, one may replace the Bregman divergence term in equation 7 with a $\psi$-Divergence and attempt deriving similar arguments for the rest of the paper.

## 3.1 Dykstra's Algorithm

In this section, we use Dykstra's algorithm (Dykstra, 1983) optimize objective equation 7. Dykstra is a method of iterative projections onto general closed convex constraint sets, which is well suited because the occupancy measure constraint is on a compact convex polytope $\Delta$.

Defining the Proximal map of a convex function $\phi$, with respect to $D_\Gamma$, as

$$\text{Prox}_\phi^{D_\Gamma}(\mu) = \arg\min_{\tilde{\mu}} D_\Gamma(\tilde{\mu}|\mu) + \phi(\tilde{\mu}),$$

for any $\mu \in \text{dom}(\Gamma)$, we present the following proposition which is the generalization of Dykstra algorithm in (Peyré, 2015):

**Proposition 3.1** (Dykstra's algorithm). [1] For iteration $l > 0$,

$$\mu^{(l)} = \text{Prox}_{\phi_{[l]_N}}^{D_\Gamma} \left( \nabla\Gamma^* \left( \nabla\Gamma(\mu^{(l-1)}) \right) + \nu^{(l-N)} \right) \tag{9}$$

$$\nu^{(l)} = \nu^{(l-N)} + \nabla\Gamma(\mu^{(l-1)}) - \nabla\Gamma(\mu^{(l)}), \tag{10}$$

with $[l]_N = \begin{cases} N & \text{if } l \bmod N = 0 \\ l \bmod N & \text{otherwise} \end{cases}$, converges to the solution of optimization equation 7, with $\mu^{(0)} = \xi$ and $\nu^{(i)} = \mathbf{0}$ for $-N + 1 \leq i \leq 0$.

---

[1]proofs and derivations are in supplementary material.

Intuitively, at step $l$, algorithm equation 9 projects $\mu^{(l-1)}$ into the convex constraint set corresponding to the function $\phi_{[l]_N}$.

**Corollary 3.2.** Taking $\Gamma = \mathcal{H}$, the iteration equation 9 is

$$\mu^{(l)} = \text{Prox}_{\phi_{[l]_N}}^{\text{KL}} \left( \mu^{(l-1)} \odot z^{(l-N)} \right), \quad z^{(l)} = z^{(l-N)} \odot \frac{\mu^{(l-1)}}{\mu^{(l)}}, \tag{11}$$

where $\odot, \frac{\cdot}{\cdot}$ are the element-wise product and division, $\mu^{(0)} = \xi$ and $z^{(i)} = \mathbf{1}$, for $-N + 1 \leq i \leq 0$.

Given probability measures $a, b$, for $\Gamma = \mathcal{H}$, $N = 2$, $\phi_1(\mu) = \delta_{\{a\}}(\mu\mathbf{1})$, $\phi_2(\mu) = \delta_{\{b\}}(\mu^T\mathbf{1})$, optimization equation 7 is entropic regularized OT problem and algorithm 11, is the well known Sinkhorn-Knopp algorithm (Cuturi, 2013). Similarly one can apply equation 11 to solve the regularized UOT problem (Chizat et al., 2017; 2019).

As aforementioned RL objectives in Section 2.2 can be viewed as instances of optimization equation 7, Dykstra's algorithm can be used to optimize the objectives. In particular, as the constraint $\phi_N(\mu) = \delta_{\{b^\mu\}}(A^\mu\mu)$ occurs in each objective, each iteration of Dykstra requires

$$\text{Prox}_{\phi_N}^{D_\Gamma}(\mu) = \arg\min_{\tilde{\mu} \in \Delta} D_\Gamma(\tilde{\mu}|\mu), \tag{12}$$

which is the Bregman projection of $\mu$ onto the set of occupancy measures $\Delta$.

Although $\mu$ (the measure from the previous step of Dykstra) does not necessarily lie inside $\Delta$, step equation 12 of Dykstra could be seen as a Bregman divergence policy optimization resulting in dual formulation over value functions. (See details of dual optimization in Supplementary Material.) This dual formulation is similar to REPS algorithm (Peters et al., 2010).

In the next section we apply Dykstra to solve unbalanced optimal transport on $\Delta$.

## 4   Distributionally-Constrained Policy Optimization

A natural constraint in policy optimization is to enforce a global action execution allotment and/or state visitation frequency. In particular, given a positive baseline measure $\eta'$, with $\eta'(a)$ being a rough execution allotment of action $a$ over whole state space, for $a \in \mathcal{A}$, we can consider $D_{\psi_1}\left(\mathbb{E}_{\rho^\pi}[\pi] \mid \eta'\right)$ as a global penalty constraint of policy $\pi$ under its natural state distribution $\rho^\pi$. Similarly, the penalty $D_{\psi_2}\left(\rho^\pi \mid \rho'\right)$ enforces a cautious or exploratory constraint on the policy behavior by avoiding or encouraging visitation of some states according to a given positive baseline measure $\rho'$ on $\mathcal{S}$.

So, given baseline measures $\rho'$ on $\mathcal{S}$ and $\eta'$ on $\mathcal{A}$, we define distributionally-constrained policy optimization:

$$\max_{\mu \in \Delta} \mathbb{E}_\mu[r] - \epsilon_1 D_{\psi_1}\left(\mu\mathbf{1} \mid \rho'\right) - \epsilon_2 D_{\psi_2}\left(\mu^T\mathbf{1} \mid \eta'\right). \tag{13}$$

When $D_{\psi_1} = D_{\psi_2} = \text{KL}$, objective equation 13 looks similar to equation 3 (considering expansion in equation 4), but they are different. In equation 13, if $\rho' = \mu'\mathbf{1}$ and $\eta' = \mu'^T\mathbf{1}$, for some baseline $\mu' \in \Delta$, then the third term is $\text{KL}\left(\mathbb{E}_{\rho^\pi}[\pi] \mid \mathbb{E}_{\rho^{\pi'}}[\pi']\right)$ which is a global constraint on center of mass of $\pi$ over the whole state space, whereas $\mathbb{E}_{\rho^\pi}[\text{KL}(\pi \mid \pi')]$ in equation 4 is a stronger constraint on closeness of policies on every single state. The bottom line is that equation 13 generally constrains the projected marginals of $\mu$ over $\mathcal{S}$ and $\mathcal{A}$, and equation 3 constrains $\mu$ element wise.

For regularization purposes in iterative policy optimization algorithm (e.g., using mirror descent), one natural choice of the state and action marginals is to take $\rho' = \sum_a \mu_{k-1}$, $\eta' = \sum_s \mu_{k-1}$ at the $k$'th iteration. In the Supplementary Material, we discuss the policy improvement and convergence of such an algorithm. Another source for the marginals $\rho', \eta'$ is the empirical visitation of states and actions sampled out of an expert policy in imitation learning.

Formulation of Objective equation 13 is in form of UOT on the set of occupancy measures. So, for applying Dykstra algorithm, we can add an entropy term $\epsilon \mathcal{H}(\mu)$ to transform equation 13 into

$$\max_{\mu \in \Delta} - \mathrm{KL}(\mu \mid \xi) - \epsilon_1 D_{\psi_1} \left( \mu \mathbf{1} \mid \rho' \right) - \epsilon_2 D_{\psi_2} \left( \mu^T \mathbf{1} \mid \eta' \right), \tag{14}$$

which means setting $N = 3$, $\phi_1 = \epsilon_1 D_{\psi_1}, \phi_2 = \epsilon_2 D_{\psi_2}, \phi_3(\mu) = \delta_{b^\mu}(A^\mu \mu), \xi = e^{r/\epsilon}$ in Objective equation 7 [2]. Hence, the algorithm equation 11 can be applied with following proximal functions:

$$\mathrm{Prox}_{\phi_1}^{\mathrm{KL}}(\mu) = \mathrm{diag} \left( \frac{\mathrm{Prox}_{\epsilon_1 D_{\psi_1}}^{\mathrm{KL}} (\mu \mathbf{1})}{\mu \mathbf{1}} \right) \mu, \tag{15}$$

$$\mathrm{Prox}_{\phi_2}^{\mathrm{KL}}(\mu) = \mu \, \mathrm{diag} \left( \frac{\mathrm{Prox}_{\epsilon_2 D_{\psi_2}}^{\mathrm{KL}} (\mu^T \mathbf{1})}{\mu^T \mathbf{1}} \right), \tag{16}$$

$$\mathrm{Prox}_{\phi_3}^{\mathrm{KL}}(\mu) = \arg \min_{\tilde{\mu} \in \Delta} \mathrm{KL}(\tilde{\mu}|\mu). \tag{17}$$

In general, for appropriate choices of $\phi_1, \phi_2$ (e.g., $D_{\psi_1} = D_{\psi_2} = \mathrm{KL}$) the proximal operators in equation 15 have closed form solutions. However, as discussed, $\phi_3$ in equation 17 has no closed form solution (see Supplementary Material). We can also consider other functions for $\phi_1, \phi_2$ in this scenario, for example, setting $\phi_2(\mu) = \delta_{\{\eta'\}}(\mu^T \mathbf{1})$, changes the problem into finding a policy $\pi$ which globally matches the distribution $\eta'$ under its natural state distribution $\rho^\pi$, i.e., $\mathbb{E}_{s \sim \rho^\pi}[\pi(a|s)] = \eta(a)$ for any $a \in \mathcal{A}$ [3].

In the next section we propose an actor-critic algorithm for large scale reinforcement learning.

### 4.1 Large Scale RL

When $\mathcal{S}, \mathcal{A}$ are large, policy optimization via Dykstra is challenging because tabular updating of $\mu(s, a)$ is time consuming or sometimes even impossible. In addition, it requires knowledge of reward function $r$ for the initial distribution $\xi$ and transition model $P$ for projection onto $\Delta$ in equation 17. Model estimation usually requires large number of state-action samples. Also, we might only have off-policy samples or in imitation learning scenarios, only access the marginals $\rho', \eta'$ through observed samples by the expert. In this section, we derive an off-policy optimization scheme with function approximation to address these problems.

Replacing the last two terms of obj. equation 13 with their convex conjugates by dual variables $u, v, Q$, we get

$$\max_{\mu} \min_{u,v,Q} \mathbb{E}_\mu \left[ r - A^{\mu*} Q - \epsilon_1 u \mathbf{1}_A^T - \epsilon_2 \mathbf{1}_S v^T \right] + \mathbb{E}_{b^\mu}[Q(s,a)] + \epsilon_1 \mathbb{E}_{\rho'}[\psi_1^*(u(s))] + \epsilon_2 \mathbb{E}_{\eta'}[\psi_2^*(v(a))], \tag{18}$$

where $A^{\mu*}$ is the convex conjugate (transpose) of $A^\mu$.

Having access to samples from a baseline $\mu' \in \Delta$, both helps regularize the objective equation 18 into an easier problem to solve, and allows off-policy policy optimization or imitation learning (Nachum & Dai, 2020). Yet, by the discussion in Section 4, regularizing with the term $D_\psi(\mu|\mu')$ in equation 18 might make marginal penalties redundant, in particular when $\rho' = \sum_a \mu'$ and $\eta' = \sum_s \mu'$.

Here, we discuss the case where $\rho' \neq \sum_a \mu'$ or $\eta' \neq \sum_s \mu'$. See the supplemental for $\rho' = \sum_a \mu'$ and $\eta' = \sum_s \mu'$. Without the loss of generality, assume marginals $\rho' \neq \sum_a \mu'$ and $\eta' \neq \sum_s \mu'$. In this case, regularizing equation 18 with $D_\psi(\mu|\mu')$ and under Fenchel duality, we get the off-policy optimization objective

$$\max_{\pi} \min_{u,v,Q} \mathbb{E}_{\mu'} \left[ \psi^* \left( r + \gamma P^\pi Q - Q - \epsilon_1 u - \epsilon_2 v \right) (s,a) \right]$$
$$+ (1-\gamma) \mathbb{E}_{\pi, p_0}[Q(s,a)] + \epsilon_1 \mathbb{E}_{\rho'}[\psi_1^*(u(s))] + \epsilon_2 \mathbb{E}_{\eta'}[\psi_2^*(v(a))], \tag{19}$$

where $u(s,a) := u(s)$ and $v(s,a) := v(a)$. Now, the first term is based on expectation of baseline $\mu'$ and can be estimated from offline samples. In a special case, when $D_{\psi_1} = D_{\psi_2} = \mathrm{KL}$ in objective equation 13 and

---

[2]Coefficients $\epsilon_1$ and $\epsilon_2$ in equation equation 14 are different from those in equation equation 13.

[3]As a constraint, $\pi \rho^\pi$ would be a feasible solution, when $\pi(a|s) = \eta'(a)$.

we take $D_\psi = \mathrm{KL}$ as well, similar derivations yield

$$\max_\pi \min_{u,v,Q} \mathcal{L}(\pi, u, v, , Q) := \log \mathbb{E}_{\mu'} \left[ \exp \left( r + \gamma P^\pi Q - Q - \epsilon_1 u - \epsilon_2 v \right)(s,a) \right] +$$
$$(1-\gamma)\mathbb{E}_{p_0,\pi}[Q(s,a)] + \epsilon_1 \log \mathbb{E}_{\rho'}[\exp(u(s))] + \epsilon_2 \log \mathbb{E}_{\eta'}[\exp(v(a))]. \tag{20}$$

In this objective note how the $\epsilon_1 u$ and $\epsilon_2 v$ are subtracted in the first term under distribution $\mu'$ and the last two term are added in the objective with respect to distributions $\rho'$ and $\eta'$ to adapt the distribution of baseline samples $\mu'$ to the constraint distributions $\rho'$ and $\eta'$.
Also, the policy evaluation problem for a fixed policy $\pi$ corresponding to this objective 20 (i.e., $\min_{u,v,Q} \mathcal{L}(u,v,,Q;\pi)$ ) corresponds to the dual form update of the Dykstra's algorithm when $\Gamma = \mathcal{H}$. This connects the Dykstra's algorithm in U(OT) literature to the minimax optimization of Reinforcement Learning here and work of (Nachum & Dai, 2020). Please see the equation 25 in proof of Dykstra's in the appendix A.

Now, gradients of $\mathcal{L}$ can be computed with respect to $u, v, Q$:

$$\nabla_u \mathcal{L} = -\epsilon_1 \mathbb{E}_{(s,a)\sim\mu'}[\mathcal{F}_{\mu'} \circ h^\pi_{u,v,Q}(s,a)\nabla u(s)] + \epsilon_1 \mathbb{E}_{s\sim\rho'}[\mathcal{F}_{\rho'} \circ u(s)\nabla u(s)], \tag{21}$$

$$\nabla_v \mathcal{L} = -\epsilon_2 \mathbb{E}_{(s,a)\sim\mu'}[\mathcal{F}_{\mu'} \circ h^\pi_{u,v,Q}(s,a)\nabla v(a)] + \epsilon_2 \mathbb{E}_{a\sim\eta'}[\mathcal{F}_{\eta'} \circ v(a)\nabla v(a)], \tag{22}$$

$$\nabla_Q \mathcal{L} = \mathbb{E}_{(s,a)\sim\mu'}[\mathcal{F}_{\mu'} \circ h^\pi_{u,v,Q}(s,a)\left(\gamma P^\pi \nabla Q - \nabla Q\right)(s,a)] + (1-\gamma)\mathbb{E}_{\substack{s\sim p_0 \\ a\sim\pi(\cdot|s)}}[\nabla Q(s,a)]. \tag{23}$$

And, the gradient with respect to policy $\pi$ is

$$\nabla_\pi \mathcal{L} = \gamma \mathbb{E}_{\substack{a'\sim\pi(\cdot|s') \\ (s,a,s')\sim\mu'}}[\mathcal{F}_{\mu'} \circ h^\pi_{u,v,Q}(s,a)Q(s',a') \cdot \nabla \log \pi(a'\mid s')] + (1-\gamma)\mathbb{E}_{\substack{s\sim p_0 \\ a\sim\pi(\cdot|s)}}[Q(s,a)\nabla \log \pi(a\mid s)], \tag{24}$$

where, $h^\pi_{u,v,Q}(s,a) := r(s,a) + \gamma P^\pi Q(s,a) - Q(s,a) - \epsilon_1 u(s) - \epsilon_2 v(a)$ and $\mathcal{F}_p \circ h(z) := \mathcal{F}_p(h)(z) = \exp(h(z))/\mathbb{E}_{\tilde{z}\sim p}[\exp(h(\tilde{z}))]$ is the softmax operator for any $h, z, p$. Note that as in the first term in objective equation 20, $h^\pi_{u,v,Q}(s,a)$ could be viewed as the adjusted temporal difference error with respect to the *adjusted reward function* $r(s,a) - \epsilon_1 u(s) - \epsilon_2 v(a)$, seeing $u(s)$ and $v(a)$ as the cost functions associated to states and actions respectively and equation 24 is similar to regular policy gradient update for this adjusted reward function.
Finally, $u, v, Q, \pi$ can be approximated by some functions (e.g., Neural networks) and one can apply primal-dual stochastic gradient algorithm on $\pi$ and $u, v, Q$. Treating $Q$ as a vector of size $\mathcal{S} \times \mathcal{A}$, algorithm 1 shows the pseudo-code for optimizing 20.

---

**Algorithm 1:** Large Scale RL when $\rho' \neq \sum_a \mu'$ **or** $\eta' \neq \sum_s \mu'$

---

**Input:** samples $\{(s_i, a_i, r_i)\}_{i=1}^n \sim \mu' \in \Delta$, $\{s'_j\}_{j=1}^n \sim \rho'$, $\{a'_j\}_{j=1}^n \sim \eta'$, $\epsilon_1$, $\epsilon_2$, learning rates $\beta_1$, $\beta_2$

Replace Expectations in equation 21,equation 22,equation 23 and equation 24 by empirical averages over samples.

$[u, v, Q]^T \leftarrow [u, v, Q]^T - \beta_1 [\nabla_u \mathcal{L}, \nabla_v \mathcal{L}, \nabla_Q \mathcal{L}]^T$

$\pi \leftarrow \pi + \beta_2 \nabla_\pi \mathcal{L}$

**Output:** $\pi$

---

See the supplemental text for the gradient derivations. Also, note that setting $u$ and $v$ to $\mathbf{0}$, in our updates, we can recover the dice-method optimization in Nachum & Dai (2020) with similar update for $Q$ in equation 23, for which the gradient calculation suffers from double sampling because of multiplication of two expectations with respect to $P^\pi$.

## 5 Demonstrations

In this section, we demonstrate the effectiveness of distributionally-constrained policy optimization with Dykstra. The purpose of our experiments is to answer how distributional penalization on $\rho'$ and $\eta'$ affect

the behavior of the policy and study the Dykstra's empirical rate of convergence. We test these questions at a granular level for a grid-world example first and next we test it for the cart-pole problem as a large RL task.

## 5.1 Grid-world example

We look into these questions on a grid world with its optimal policy out of equation 6 shown in Fig. 1. Due to the simplicity, this environment is suitable for studying the effect of distributional constraints on the policy behavior. For the sake of focusing on the effects of distributional constraints, we set the coefficient of the entropy term fairly low ($\epsilon = .01$) in optimizations equation 6 and equation 14. [4]

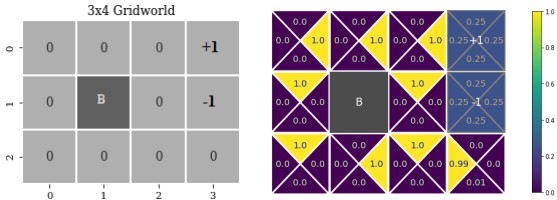

Figure 1: **Left**: The grid world MDP. The reward entering each state is shown on the grid. There is a block at state $(1, 1)$ and $\mathcal{A} = \{$up, down, left, right$\}$. Every action succeeds to its intended effect with probability 0.8 and fails to each perpendicular direction with probability 0.1. Hitting the walls means stay and $\gamma = .95$. An episode starts in $(2, 0)$ (bottom left corner) and any action in $(0, 3)$ or $(1, 3)$ transitions into $(2, 0)$. **Right**: Optimal Policy out of optimization equation 6 for $\epsilon = .01$.

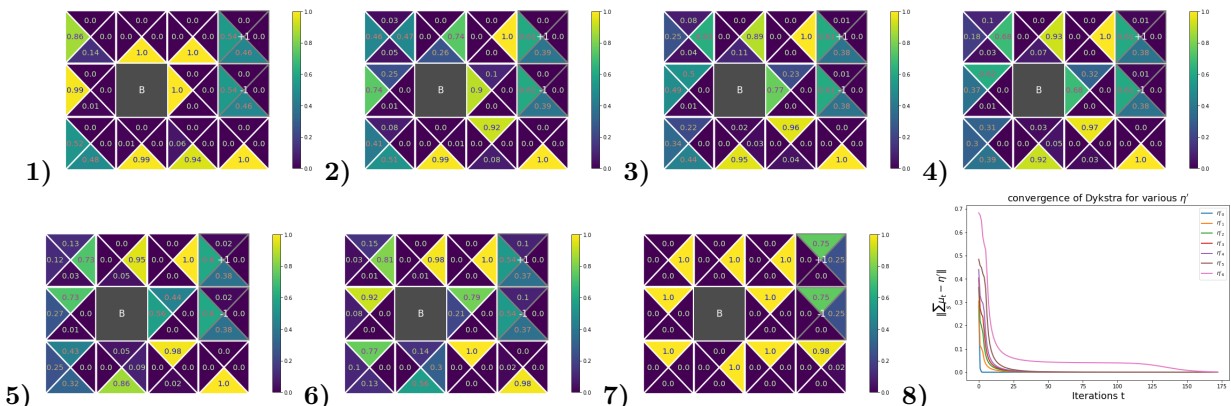

Figure 2: **(1-7)**: Optimal policy out of Dykstra's algorithm, with $\eta_i' = [\alpha, .5 - \alpha, .5 - \alpha, \alpha]$ for $\alpha_i \in \{e^{-10}, .1, .2, .25, .3, .4, .5 - e^{-10}\}$ for $i \in \{1, \ldots, 7\}$. **(8)**: corresponding convergence of the Dykstra in (1-7), horizontal axis is the number of iterations $t$ and the vertical axis is $\|\sum_s \mu_t - \eta_i'\|$.

We first observe the independent effect of $\eta'$ on the policy, by setting $\epsilon_1 = 0$. We use, $\delta_{\eta'}(\mu^T \mathbf{1})$ instead of $D_{\psi_1}(\mu^T \mathbf{1} \mid \eta')$ as an extreme case when $\epsilon_2 \to \infty$ to focus on the role $\eta'$. Figure 2(1-7), shows differences among policies when the marginal on actions shifts from a distribution where only equiprobable actions *down* and *left* are allowed ($\eta_1' = (0, 0.5, 0.5, 0)$) towards the case where only *up* and *right* are permitted with equal probability ($\eta_7' = (0.5, 0, 0, 0.5)$). [5]

In Figure 2(1), under $\eta_1'$, *down* is the optimal action in state $(0, 2)$ because, this is the only way to get $+1$ reward (with luck). In 2(2), which changes to a .1 probability on *right*, the policy eliminates the reliance on change by switching state $(0, 2)$ to right.

---

[4]Supplementary Material provides the numerical settings in implementation of Dykstra.

[5]The optimal policies in this section aren't necessarily deterministic (even though $\epsilon$ is set to be very small), because of the constraint $\delta_{\eta'}(\mu(a))$. In general, the policies out of equation 14 are not necessarily deterministic either because of the nonlinear objective.

Note that the optimal policy in Figure 1(left) does not include a *down* move. When *down* is forced to have non-zero probability, Figures 2(1-6), the policy assigns it to state $(2, 3)$, towards the case where only *up* and *right* are permitted with equal probability ($\eta'_7 = (0.5, 0, 0, 0.5)$).

Figures 2(7) shows the case where only *up* and *right* are allowed. In state $(2, 3)$, this creates a quandary. *Right* is unlikely to incur the $-1$ penalty, but will never allow escape from this state. For this reason, the policy selects *up*, which with high probability will incur to the $-1$ penalty, but has some probability of escape toward the $+1$ reward.

Figure 2(8), depicts the convergence of Dykstra towards various $\eta'$. Notably, in all cases the algorithm converges, and the rate of convergence increases following the order of the subfigures.

Next, we test the extreme effect of constraints on the state marginals on the policy via various $\rho'$, by setting $\epsilon_2 = 0$ and $\epsilon_1$ very high. We study the policy when $\rho'(s) = .9$ for a single state $s$, and uniform distribution of $.1$ on the rest of states other than $s$. Figure 3(1-3) shows the policies when $s \in \{(0, 2), (1, 2), (2, 3)\}$. Hitting the wall seems to be viable strategy to stay and increase the visitation frequency of each of these states. Figure 3(4) depicts the the convergence of Dykstra's algorithm towards various $\rho'$. As shown, the error never gets to zero. This is because by setting $\epsilon_1 \to \infty$, the objective is just to find an occupancy measure with closest state marginal to $\rho'$ and $D_{\psi_1}(\mu\mathbf{1} \mid \rho')$ can never be zero if $\rho'$ is not from a valid occupancy measure.

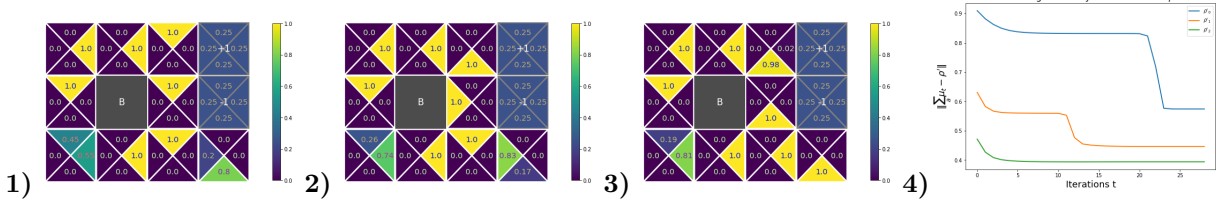

Figure 3: **(1-3)**: Optimal policy out Dykstra, when $\rho'_i(s) = .9$ for $s \in \{(0, 2), (1, 2), (2, 3)\}$ and uniform distribution of $.1$ on the rest of states other than $s_i$ for $i \in \{1, 2, 3\}$. **(4)**: corresponding convergence of the Dykstra in (1-3), horizontal axis is the number of iterations $t$ and the vertical axis is $\|\sum_a \mu_t - \rho'_i\|$.

We also test how imitating a policy with distributional constraints affects the learned policy. For this purpose, we create a new environment with reward $-10$ at state $(1, 3)$. The optimal risk-averse policy $\pi_1$ out of this environment is shown in Figure 4(1). Let $\eta'_1 = \sum_s \mu^{\pi_1}$ be the action marginal corresponding to $\pi_1$. Now consider the RL problem with reward of $-1$ in state $(1, 3)$ constrained by $\eta'_1$. Figure 4(2) shows the resulting policy $\pi_2$. Notice that $\mu^{\pi_2}$ achieves the action marginal distribution $\eta'_1$, however, $\pi_2$ is quite different from $\pi_1$, since the unconstrained optimal policy for the environment with reward of $-1$ at state $(1, 3)$ is more risk neutral. In contrast, distributionally constraining $\rho^{\pi_1} = \sum_a \mu^{\pi_1}$ (as in previous experiments) results in the same policy of $\pi_1$ as in Figure 4(1). The differences are mostly in states $(0, 3)$ and $(1, 3)$, where actions can be freely chosen (but not their visitation frequency) and contribution of state $(2, 3)$ which has a lower visitation probability. Consider constraints on both $\eta'$ and $\rho'$. As explained earlier, the policy will then get as

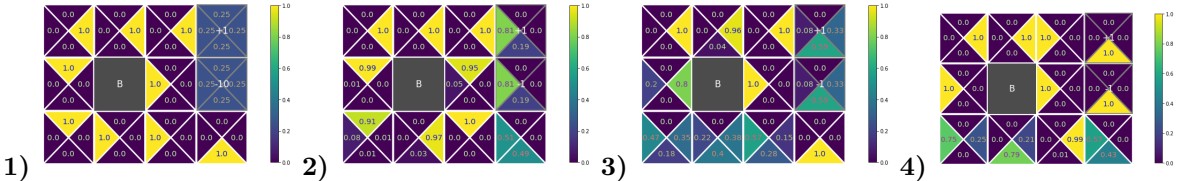

Figure 4: **(1)**: Risk averse optimal policy learned by changing $-1$ to $-10$ in state $(1, 3)$. **(2)**: Optimal policy using $\eta'$ out of risk-averse policy as a distributional constraint. **(3)**: Optimal policy with $\eta' = [e^{-10}, \alpha, \alpha, \alpha]$, $\alpha = (1 - e^{-10})/3$. **(4)**: Policy with the same $\eta'$ and $\rho'_1$.

close as possible to $\rho'$ while satisfying the action distribution $\eta'$. Figure 4-(3) shows the optimal policy for $\eta'$ with no constraint on $\rho'$. $\eta'$ is a distribution where no *up* is allowed and the other three actions equiprobable.

Figure 4-(4) depicts the policy under constraints on both $\eta'$ and $\rho'$ when $\rho'$ is the same distribution in Figure 3(1). The leftmost column and top row of this policy leads to (0,2) but in an attempt to satisfy $\rho'$, the policy goes back to the *left*.

## 5.2   Cart-pole example

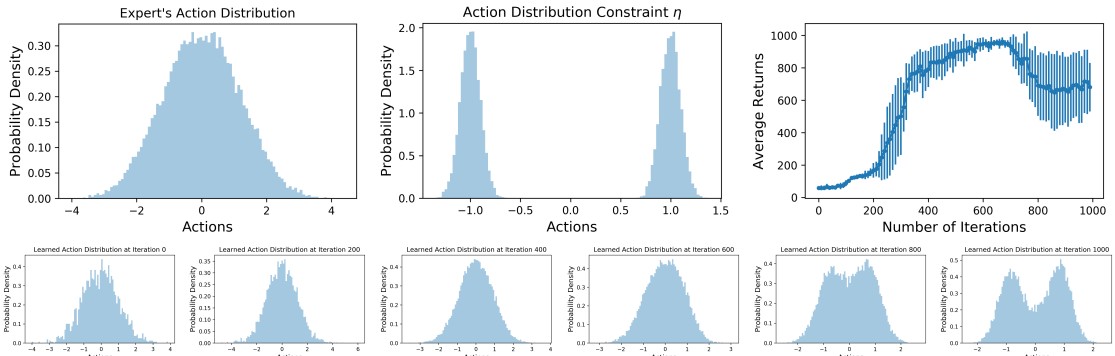

Figure 5: (Top row-left to right):Action distribution of expert policy, desired distribution $\eta'$, accumulated reward during learning. (bottom row) Evolution of action distribution in Algorithm 1.

We also perform an ablation study on the Algorithm 1 (see Appendix) in an imitation learning scenario on the cart-pole problem. In this test, the expert policy is learned by the policy gradient algorithm and we imitate the learned policy (assuming $r(s, a) = 0$) with Algorithm 1 enforcing a given global action distribution $\eta'$ post-learning. In this setting the expert global action distribution out of a random run is shown in the top-left corner of Figure 5 and $\eta'$ is bimodal distribution forcing the cart to choose values around $-1$ and $+1$ (Top-middle figure). The top-right figure shows the average return of the learned policy (under 3 seeds). The second row of figure shows the shaping and splitting of the action marginal under primal-dual iterations of Algorithm 1.

## 6   Related Works

• **Objectives and Constraints in Reinforcement Learning.** Posing policy optimization as a constrained linear programming on the set of occupancy measures has been long studied (Puterman, 2014). Recent works have expanded linear programming view through a more general convex optimization framework. For example, (Neu et al., 2017) unifies policy optimization algorithms in literature for entropy regularized average reward objectives. (Nachum et al., 2019b;a; Nachum & Dai, 2020) propose entropy-regularized policy optimizations under Fenchel duality to incorporate the occupancy measure constraint. Unlike these works, we looked at policy optimization from an OT point of view. To do so, we proposed a structured general objective based on Bregman divergence that allows considering relaxations of entropy regularization using marginals. (Zhang et al., 2020b) studies a general concave objective in RL and proposes a variational Monte Carlo algorithm using the Fenchel dual of the objective function. Similar to these works we take advantage of Fenchel duality. However, other than different view point and structured objective, our work differs in solving the optimization by breaking the objective using Dykstra's algorithm. (Zhang et al., 2020a) proposes various *caution penalty functions* as the RL objective and a primal-dual approach to solve the optimization. One of these objectives is a penalty on $\mathrm{KL}(\cdot|\rho')$, which is a part of our proposed unbalanced formulation. Other than the formulation, in this work, we focused on distributional penalty on global action execution which, to the best our knowledge, has not been studied before.

In constrained MDPs, a second reward $c(s, a) < 0$ is used to define a constrained value function $C^\pi$ (Altman, 1999; Geibel, 2006). Here $C^\pi(s) = \mathbb{E}_\pi \left[ \sum_{t=0}^\infty \gamma^t c(s, a) \right]$ and the constraint is in form of $\mathbb{E}_\pi \left[ C^\pi(s) \right] > A$ $(*)$, where $A$ is a constant. Thus considering $c(s, a)$ as the cost for taking action $a$ at state $s$, constrained MDP optimizes the policy with a fixed upper bound for the expected cost. Rather than introducing a fixed scalar

restriction (A), our formulation allows distributional constraints over both the action and state spaces (i.e. $\rho'$ and $\eta'$). The source of these distributional constraints may vary from an expert policy to the environmental costs and we can apply them via penalty functions. In special cases, when an action can be identified by its individual cost, constraint $(*)$ on expected cost can be viewed as a special case of marginal constraint on $\eta'$. For instance, in the grid world of Fig. 1, if the cost for *up* and *right* is significantly higher than *down* and *left*, then limited budget (small expected cost) is essentially equivalent to having a marginal constraint $\eta'_1$ supported on *down* and *left*. However, in general, when $c(s, a)$ varies for state per action, the expected cost does not provide much guidance over global usage of actions or visitation of states as our formulation does.

• **Reinforcement Learning and Optimal Transport** OT in terms of Wasserstein distance has been proposed in the RL literarture. (Zhang et al., 2018) views policy optimization as Wasserstein gradient flow on the space of policies. (Pacchiano et al., 2020) defines behavioral embedding maps on the space of trajectories and uses an approximation of Wasserstein distance between measures on embedding space as regularization for policy optimization. Marginals of occupancy measures can be viewed as embeddings via state/action distribution extracting maps. Our work defines an additive structure on these embedding functionals which is broken into Bregman projections using Dykstra.

• **Imitation Learning and Optimal Transport** In the imitation learning literature, (Xiao et al., 2019) proposed an adversarial inverse reinforcement learning method which minimizes the Wasserstein distance to the occupancy measure of the expert policy using the dual formulation of optimal transport. (Dadashi et al., 2020) minimized the primal problem of Wasserstein minimization and (Papagiannis & Li, 2020) minimize the *Sinkhorn Divergence* to the expert's occupancy measure. These works are fundamentally different from our approach as we are not solving the inverse RL problem and we view RL itself as a problem of stochastic assignment of actions to states. The type of distributional constraints via unbalanced optimal transport proposed in our work can be considered as relaxation of the idea of matching expert policy occupancy measures. We consider matching the distribution of global action executions and state visitations of the expert policy.

• **Related works in Optimal Transport** It is not the first time (U) OT is considered constrained. Martingale optimal transport imposes an extra constraint on the mean of the coupling (Beiglböck et al., 2013) and using entropic regularization, Dykstra can be applied (Henry-Labordère, 2013).

• **Other Settings** In UOT formulation of equation 13 we used penalty $D_\psi$. One can apply other functions like the indicator function to enforce constraints on marginals like $\delta_{\eta'}(\mu^T \mathbf{1})$ as discussed in Sec. 4. However, using the constraint $\delta_{\rho'}(\mu \mathbf{1})$ in equation 13 could be problematic as it can easily be incompatible with the occupancy measure constraint. If $\rho'$ is not coming form a policy, then the optimization is infeasible. Despite this, setting $\epsilon_2 = 0$, $\rho' = \rho^{\pi_{k-1}}$ in equation 13 for the $k$'th iteration of an iterative policy optimization algorithm, equation 13 results in objective similar to **TRPO** (Schulman et al., 2015).

## 7 Conclusion

We have introduced distributionally-constrained policy optimization via unbalanced optimal transport. Extending prior work, we recast RL as a problem of unbalanced optimal transport via minimization of an objective with a Bregman divergence which is optimized through Dykstra's algorithm. We illustrate the theoretical approach through the convergence and policies resulting from marginal constraints on $\eta'$ and $\rho'$ both individually and together. The result unifies different perspectives on RL and naturally allows incorporation of a wide array of realistic constraints on policies. As discussed, we developed the general objectives using Bregman divergence $D_\Gamma$ and $\psi$-divergence $D_\psi$ and experimented on the case with $D_\Gamma = D_\psi = \text{KL}$, in this paper. It is an interesting direction for future work to test for other various $\Gamma$'s and $\psi$'s.

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

## A    Fenchel Dual and Proofs in Section 3

For any function $f : \mathrm{Dom}(f) \to \mathbb{R}$, its convex conjugate (or Fenchel dual) $f^*$ is defined as $f^*(y) = \max_x \langle x, y \rangle - f(x)$. If $f$ is proper, convex and lower-semi continuous then $f^*$ has the same properties and one can write $f(x) = \max_y \langle x, y \rangle - f^*(y)$. If $f$ is strictly convex and smooth on $\mathrm{int}(\mathrm{dom} f)$, then $\nabla f$ and $\nabla f^*$ are bijective maps between $\mathrm{int}(\mathrm{dom} f^*)$ and $\mathrm{int}(\mathrm{dom} f)$, i.e., $\nabla f^* = \nabla f^{-1}$. It is easy to verify that

- For $f(x) = \delta_a(x)$, $f^*(y) = \langle a, y \rangle$ .

- Consider $f$ with $\mathrm{Dom}(f) = \mathcal{M}(Z)$, where $Z$ is an underlying space. For a fixed $p \in \mathcal{M}(Z)$, if $f(x) = D_\psi(x|p)$, $f^*(y) = \mathbb{E}_p(\psi^*(y))$. Also, if $\psi(z) = z \log z$, then $f(x) = \mathrm{KL}(x|p)$ and $f^*(y) = \log \mathbb{E}_p(\exp(y))$.

*Proof of Proposition 3.1.* Under condition equation 8, the Fenchel-Legendre duality holds and the solution of optimization equation 7 can be recovered via

$$\max_{u_1,\cdots,u_N} - \sum_{i=1}^{N} \phi_i^*(u_i) - D_\Gamma^*\left(-\sum_{i=1}^{N} u_i | \xi\right) = \max_{u_1,\cdots,u_N} - \sum_{i=1}^{N} \phi_i^*(u_i) - \Gamma^*\left(\nabla\Gamma(\xi) - \sum_{i=1}^{N} u_i\right) - \langle\nabla\Gamma(\xi), \xi\rangle + \Gamma(\xi)$$

(25)

with the primal-dual relationship

$$\mu = \nabla\Gamma^*\left(-\sum_{i=1}^{N} u_i\right).$$

Applying coordinate descent on equation 25, with initial condition $(u_1^{(0)}, \cdots, u_N^{(0)}) = (0, \cdots, 0)$, and setting $i = [l]_N$, $J = \{1, \cdots, N\} \setminus \{i\}$, at $l > 0$ we get the iteration

$$u_i^{(l)} = \arg\max_{u_i} -\phi_i^*(u_i) - \Gamma^*(q - u_i),$$

$$u_j^{(l)} = u_j^{(l-1)}, \forall j \in J,$$

(26)

where $q = \nabla\Gamma(\xi) - \sum_{j=1}^{J} u_j^{(l-1)}$. The primal problem of optimization in equation 26 is

$$\arg\min_{\mu_i} \Gamma(\mu_i) - \langle q, \mu_i \rangle + \phi_i(\pi_i) = \arg\min_{\mu_i} D_\Gamma\left(\mu_i | \nabla\Gamma^*(q)\right) + \phi_i(\mu_i) + \text{const} = \text{Prox}_{\phi_i}^{D_\Gamma}(\nabla\Gamma^*(q)).$$

Hence, under the relation $\mu_i = \nabla\Gamma^*(q - u_i)$, we can rewrite equation 26 as

$$u_i^{(l)} = q - \nabla\Gamma\left(\text{Prox}_{\phi_i}^{D_\Gamma}(\nabla\Gamma^*(q))\right).$$

Hence, for $i = [l]_N$, we have $\mu_i^{(l)} = \nabla\Gamma^*(q - u_i^{(l)})$

$$\mu^{(l)} = \nabla\Gamma^* \circ \nabla\Gamma\left(\text{Prox}_{\phi_i}^{D_\Gamma}(\nabla\Gamma^*(q))\right) = \text{Prox}_{\phi_i}^{D_\Gamma}(\nabla\Gamma^*(q)) = \text{Prox}_{\phi_i}^{D_\Gamma}\left(\nabla\Gamma^*\left(q - u_i^{(l-N)} + u_i^{(l-N)}\right)\right)$$

$$= \text{Prox}_{\phi_i}^{D_\Gamma}\left(\nabla\Gamma^*\left(\nabla\Gamma(\mu^{(l-1)}) + u_i^{(l-N)}\right)\right).$$

Calculating the difference $\nabla\Gamma(\mu^{(l-1)}) - \nabla\Gamma(\mu^{(l)})$ and change of variable $\nu^{(l)} = -u_{[l]_N}^{(l)}$, ends the proof. $\quad\square$

*Proof of Corollary 3.2.* Peyré (2015) Setting $\Gamma = \mathcal{H}$, then $\nabla\Gamma = \log$ and $\nabla\Gamma^* = \exp$. Also, $D_\Gamma = \text{KL}$ and by change of variable $z^{(l)} = \nabla\Gamma(v^{(l)})$, the corollary follows. $\quad\square$

## B  Derivations in Section 4

Here we derive the Proximal operators in section equation 4.

### B.1  Proximal Operator Calculations when $D_\Gamma = \text{KL}$

When $D_\Gamma = \text{KL}$ (i.e., $\Gamma = \mathcal{H}$), for $\phi_3(\mu) = \delta_{b^\mu}(A^\mu \mu)$ we have:

$$\text{Prox}_{\phi_3}^{\text{KL}}(\mu) = \arg\min_{\tilde{\mu} \in \Delta} \text{KL}(\tilde{\mu}|\mu),$$

(27)

letting $V, \lambda$ to be dual variables, by the definition of $\Delta$ in lemma 2.1, the Lagrangian of equation 27 is

$$\text{KL}(\tilde{\mu}|\mu) - \gamma \sum_{s,s',a'} P(s|s',a')\mu(s',a')V(s) - (1-\gamma)\sum_s p_0(s)V(s) + \sum_{s,a} \mu(s,a)V(s) + \lambda\left(\sum_{s,a}\mu(s,a) - 1\right).$$

The derivative of the Lagrangian with respect to $\tilde{\mu}(s,a)$ for equation 27 is

$$\log(\tilde{\mu}(s,a)/\mu(s,a)) - \gamma\sum_{s'} P(s'|s,a)V(s') + V(s) + \lambda = 0.$$

So, the optimal solution is

$$\tilde{\mu}(s,a) = e^{-\lambda}\mu(s,a)\exp(\gamma\sum_{s'} P(s'|s,a)V(s') - V(s)).$$

Since $\sum_{s,a} \tilde{\mu}(s,a) = 1$, we have

$$\lambda = \log \sum_{s,a} \mu(s,a) \exp\left(\gamma \sum_{s'} P(s'|s,a)V(s') - V(s)\right),$$

then

$$\tilde{\mu}(s,a) = \frac{\mu(s,a) \exp\left(\gamma \sum_{s'} P(s'|s,a)V(s') - V(s)\right)}{\sum_{s,a} \mu(s,a) \exp(\gamma \sum_{s'} P(s'|s,a)V(s') - V(s))}, \tag{28}$$

where $V$ is the solution of the dual problem

$$\min_{\lambda,V} \lambda + (1-\gamma) \sum_s p_0(s)V(s) =$$
$$\min_V \log \sum_{s,a} \mu(s,a) \exp(\gamma \sum_{s'} P(s'|s,a)V(s') - V(s)) + (1-\gamma) \sum_s p_0(s)V(s). \tag{29}$$

By solving optimization equation 29, we can recover optimal $\tilde{\mu}$ by equation equation 28.

For other proximal operators we need the following lemma:

**Lemma B.1.** Peyré (2015) For any convex function $h$:

(i) For any $\phi(\mu) = h(\mu\mathbf{1})$

$$\text{Prox}_\phi^{\text{KL}}(\mu) = \text{diag}\left(\frac{\text{Prox}_h^{\text{KL}}(\mu\mathbf{1})}{\mu\mathbf{1}}\right)\mu,$$

(ii) and, if $\phi(\mu) = h(\mu^T\mathbf{1})$

$$\text{Prox}_\phi^{\text{KL}}(\mu) = \mu \, \text{diag}\left(\frac{\text{Prox}_h^{\text{KL}}(\mu^T\mathbf{1})}{\mu^T\mathbf{1}}\right).$$

*Proof.* Let

$$\tilde{\mu} := \text{Prox}_\phi^{\text{KL}}(\mu) = \arg\min_{\tilde{\mu}} \text{KL}(\tilde{\mu}|\mu) + h(\tilde{\mu}\mathbf{1}),$$

hence, by the first order condition, at the optimal $\tilde{\mu}$, there exists $Z \in \partial h(\tilde{\mu}\mathbf{1})$ such that we have $\log\frac{\tilde{\mu}}{\mu} + Z = 0$. Therefore,

$$\tilde{\mu} = \text{diag}\left(e^{-Z}\right)\mu. \tag{30}$$

Similarly, considering the first order condition corresponding to $\tilde{u} := \text{Prox}_h^{\text{KL}}(\mu\mathbf{1})$, we get $\tilde{u} = \text{diag}(e^{-Z})\mu\mathbf{1}$, and combining this with equation 30 proves part (i). Transposing (i), results in (ii). □

Using this lemma, we calculate some of the proximal operators we used in Section 4 and 5.

- Let $\rho := \mu\mathbf{1}$. Given $\rho' \in \mathcal{M}(\mathcal{S})$, when $\phi_1(\mu) = \epsilon_1 D_{\psi_1}(\rho|\rho') = \epsilon_1 \text{KL}(\rho|\rho')$ ,

$$\text{Prox}_{\epsilon_1 \text{KL}}^{\text{KL}}(\rho) = \arg\min_{\tilde{\rho}} \text{KL}(\tilde{\rho}|\rho) + \epsilon_1 \text{KL}(\tilde{\rho}|\rho'),$$

then, by the first order condition, we have

$$\log\frac{\tilde{\rho}}{\rho} + \epsilon_1 \log\frac{\tilde{\rho}}{\rho'} = 0,$$

and $\text{Prox}_{\epsilon_1 \text{KL}}^{\text{KL}}(\mu\mathbf{1}) = (\rho\rho'^{\epsilon_1})^{1/(1+\epsilon_1)}$. Applying lemma B.1(i), gives the proximal operator for $\phi_1(\mu)$.

- When $\phi_1(\mu) = \delta_{\rho'}(\rho)$, then $\text{Prox}_{\phi_1}^{\text{KL}}(\rho) = \rho'$ and by applying lemma B.1(i), we get

$$\text{Prox}_{\phi_1}^{\text{KL}}(\mu) = \text{diag}\left(\frac{\rho'}{\mu\mathbf{1}}\right)\mu.$$

- Similarly, given $\eta' \in \mathcal{M}(\mathcal{A})$. If $\eta := \mu^T \mathbf{1}$ and $\phi_2 = \epsilon_2 D_{\psi_2}(\eta|\eta') = \epsilon_2 \mathrm{KL}(\eta|\eta')$,

$$\mathrm{Prox}^{\mathrm{KL}}_{\epsilon_2 \mathrm{KL}}(\eta) = \arg\min_{\tilde{\eta}} \mathrm{KL}(\tilde{\eta}|\eta) + \epsilon_2 \mathrm{KL}(\tilde{\eta}|\eta'),$$

and $\mathrm{Prox}^{\mathrm{KL}}_{\epsilon_2 \mathrm{KL}}(\mu^T \mathbf{1}) = (\eta\eta'^{\epsilon_2})^{1/(1+\epsilon_2)}$. Applying lemma B.1(ii) results in the proximal operator for $\phi_2$.

- Also, when $\phi_2(\mu) = \delta_{\eta'}(\mu^T \mathbf{1})$, by lemma B.1(ii), we get

$$\mathrm{Prox}^{\mathrm{KL}}_{\phi_2}(\mu) = \mu \, \mathrm{diag}\left(\frac{\eta'}{\mu^T \mathbf{1}}\right).$$

## B.2 Convergence of the iterative policy optimization for problems equation 13 and equation 14

We now show the convergence and policy improvement of optimization equation 13, and equation 14 in an iterative policy optimization scenario.

**Proposition B.2.** Let $(\mu_k)_{k\in\mathbb{N}}$ be a sequence of occupancy measures such that for each $k \geq 1$, $\mu_k$ is the solution to the optimization equation 13 with $\mu' = \mu_{k-1}$, then $\lim_{k\to\infty} \mu_k = \mu^*$ is the solution to $\max_{\mu\in\Delta} \mathbb{E}_\mu[r]$.

*Proof.* Let $\mu_0 \in \Delta$ such that $\mu_0(s,a) > 0$ for all $(s,a)$. Let $\mu_k = \Psi(\mu_{k-1})$ for all $k > 0$ where $\Psi(\mu')$ is the solution to equation 14. Let $\Theta(\mu) = \mathbb{E}_\mu[r]$, since $\mu_{k-1}$ is a feasible point for optimization equation 13, we have monotonic improvement on the discounted accumulated rewards as

$$\Theta(\mu_{k-1}) \leq \max_{\mu\in\Delta} \Theta(\mu) - \epsilon_1 D_{\psi_1}(\mu\mathbf{1}|\mu_{k-1}\mathbf{1}) - \epsilon_2 D_{\psi_2}(\mu^T\mathbf{1}|\mu_{k-1}^T\mathbf{1}) \leq \Theta(\mu_k), \tag{31}$$

with equality achieved only if $\mu_{k-1} = \mu_k = \mu^*$.

$\Theta(\mu) - \epsilon_1 D_{\psi_1}(\sum_a \mu | \sum_a \mu') - \epsilon_2 D_{\psi_2}(\sum_s \mu | \sum_s \mu')$ is strictly concave on $\mu$ and smooth on $\mu'$. Thus $\Psi(\mu')$ is continuous and so is $\Theta(\Psi(\mu'))$.

Since we assumed $\mathcal{S}, \mathcal{A}$ are finite, from monotone convergence theorem, we can conclude that $\lim_{k\to\infty} \Theta(\mu_k) =: \theta_\infty$ exists. Then we prove the theorem by contradiction. Suppose that $\theta_\infty < \Theta(\mu^*) = \max_{\Delta_\eta} \Theta$, then $\Theta^{-1}(\theta_\infty)$ is closed and bounded, thus compact in $\Delta$, therefore there exists a $c > 0$ such that $\Theta(\Psi(\mu)) - \Theta(\mu) \geq c$ on $\Theta^{-1}(\theta_\infty)$. By continuity and compactness we may choose a $\delta < c/2$ small enough such that $\Theta(\Psi(\mu)) - \Theta(\mu) > c/2$ on $\Theta^{-1}([\theta_\infty - \delta, \theta_\infty])$. There is such a $\delta(\mu) > 0$ for every $\mu \in \Theta^{-1}(\theta_\infty)$ and we can choose $\delta = \min \delta(\mu) > 0$ by compactness.

Since $\lim_{k\to\infty} \Theta(\mu_k) = \theta_\infty$, there exists an $n > 0$ such that $\Theta(\mu_n) \in [\theta_\infty - \delta, \theta_\infty]$, thus $\Theta(\mu_{n+1}) = \Theta(\Psi(\mu_n)) > \Theta(\mu_n) + c/2 \geq \theta_\infty - \delta + c/2 > \theta_\infty$. This is an contradiction to the assumption that $\Theta(\mu_k)$ converges to $\theta_\infty$ increasingly.

Therefore, $\theta_\infty = \Theta(\mu^*)$, and from strict convexity of $\Theta$, we have $\Theta^{-1}(\theta_\infty) = \{\mu^*\}$. So $\lim_{k\to\infty} \mu_k = \mu^*$. $\square$

Similarly, for the optimization equation 14:

$$\max_{\mu\in\Delta} -\mathrm{KL}(\mu \mid \xi) - \epsilon_1 D_{\psi_1}(\mu\mathbf{1} \mid \rho') - \epsilon_2 D_{\psi_2}(\mu^T\mathbf{1} \mid \eta'),$$

we can have the following proposition:

**Proposition B.3.** Let $(\mu_k)_{k\in\mathbb{N}}$ be a sequence of occupancy measures such that for each $k \geq 1$, $\mu_k$ is the solution to the optimization problem equation 14 with $\mu' = \mu_{k-1}$, then $\lim_{k\to\infty} \mu_k = \mu^*$ is the solution to $\max_{\mu\in\Delta} -\mathrm{KL}(\mu \mid \xi)$.

## C  Large Scale Algorithm

**C.1**  $\rho' = \sum_a \mu'$ **and** $\eta' = \sum_s \mu'$

Following the approach in Sutton et al. (2016); Nachum & Dai (2020), assuming $\pi$ is known, with the change of variable $\zeta(s,a) := \frac{\mu}{\mu'}(s,a)$, we can rewrite equation 18 with importance sampling weights as a policy evaluation problem

$$\min_{u,v,Q} \max_{\zeta} \mathcal{L}(u,v,Q,\zeta;\pi) := \mathbb{E}_{\mu'} \left[ \zeta(s,a)\left(r + \gamma P^\pi Q - Q - \epsilon_1 u - \epsilon_2 v\right)(s,a) \right]$$
$$+ (1-\gamma)\mathbb{E}_{p_0,\pi}[Q(s,a)] + \epsilon_1 \mathbb{E}_{\rho'}[\psi_1^*(u(s))] + \epsilon_2 \mathbb{E}_{\eta'}[\psi_2^*(v(a))]. \tag{32}$$

Given the optimized $Q,\zeta$, the gradient with respect to $\pi$ is

$$\nabla_\pi \mathcal{L}(u,v,Q,\zeta,\pi) = \mathbb{E}_{(s,a)\sim\mu'}\left[ \zeta(s,a)Q(s,a)\nabla \log \pi(a|s) \right]. \tag{33}$$

The gradients with respect to $u,v,Q,\zeta$ are as follows:

$$\nabla_u \mathcal{L}(u,v,Q,\zeta;\pi) = -\epsilon_1 \mathbb{E}_{\mu'}\left[ \zeta(s,a)\nabla u(s) \right] + \epsilon_1 \mathbb{E}_{\rho'}[\nabla_u \psi_1^*(u(s))], \tag{34}$$

$$\nabla_v \mathcal{L}(u,v,Q,\zeta;\pi) = -\epsilon_2 \mathbb{E}_{\mu'}\left[ \zeta(s,a)\nabla v(a) \right] + \epsilon_2 \mathbb{E}_{\eta'}[\nabla_v \psi_2^*(v(a))], \tag{35}$$

$$\nabla_Q \mathcal{L}(u,v,Q,\zeta;\pi) = \mathbb{E}_{\mu'}\left[ (\zeta + \gamma P^\pi \nabla Q - \nabla Q)(s,a) \right] + (1-\gamma)\mathbb{E}_{p_0,\pi}[\nabla Q(s,a)], \tag{36}$$

$$\nabla_\zeta \mathcal{L}(u,v,Q,\zeta;\pi) = \mathbb{E}_{\mu'}\left[ \nabla \zeta(s,a) h_{u,v,Q}^\pi(s,a) \right]. \tag{37}$$

Wrapping $\max_\pi$ around equation 32 gives the off-policy optimization. Given optimized $Q,\zeta$, the gradient with respect to $\pi$ is

$$\nabla_\pi \mathcal{L}(u,v,Q,\zeta,\pi) = \mathbb{E}_{(s,a)\sim\mu'}\left[ \zeta(s,a)Q(s,a)\nabla \log \pi(a|s) \right] \tag{38}$$

## D  Gradient Derivations

Here we derive optimization equation 18. Let's fix policy $\pi$, then following the policy optimization approach in Nachum & Dai (2020), one might define the policy evaluation problem for $\pi$ as $\max_\mu h(\mu) - \delta_{b^\pi}(A^\pi \mu)$ for any arbitrary concave function $h$ as the problem is over-constrained and $\mu$ is the unique solution of $A^\pi \mu = b^\pi$. So we define the policy evaluation problem for a fixed $\pi$ corresponding to equation 13 as

$$\max_\mu \mathbb{E}_\mu[r] - \epsilon_1 D_{\psi_1}(\mu\mathbf{1} \mid \rho') - \epsilon_2 D_{\psi_2}(\mu^T\mathbf{1} \mid \eta') - \delta_{b^\pi}(A^\pi \mu)$$

$$\overset{(a)}{=} \max_\mu \min_{u,v,Q} \langle\mu, r\rangle - \epsilon_1 \langle\mu\mathbf{1}, u\rangle + \epsilon_1 \mathbb{E}_{\rho'}[\psi_1^*(u(s))] - \epsilon_2 \langle\mu^T\mathbf{1}, v\rangle + \epsilon_2 \mathbb{E}_{\eta'}[\psi_2^*(v(a))] - \langle A^\pi \mu, Q\rangle + \mathbb{E}_{b^\pi}[Q(s,a)]$$

$$= \max_\mu \min_{u,v,Q} \langle\mu, r - A^{\pi*}Q - \epsilon_1 u\mathbf{1}_A^T - \epsilon_2 \mathbf{1}_S v^T\rangle + \mathbb{E}_{b^\pi}[Q(s,a)] + \epsilon_1 \mathbb{E}_{\rho'}[\psi_1^*(u(s))] + \epsilon_2 \mathbb{E}_{\eta'}[\psi_2^*(v(a))], \tag{39}$$

where (a) is obtained by replacing the last three terms by their convex conjugates and $A^{\pi*}$ is transpose of $A^\pi$. This gives optimization equation 18. If we regularize objective equation 18 with $D_\psi(\mu \mid \mu')$ regularization, under Fenchel duality we get

$$\max_\mu \min_{u,v,Q} \langle\mu, r - A^{\pi*}Q - \epsilon_1 u\mathbf{1}_A^T - \epsilon_2 \mathbf{1}_S v^T\rangle - D_\psi(\mu \mid \mu') + \mathbb{E}_{b^\pi}[Q(s,a)] + \epsilon_1 \mathbb{E}_{\rho'}[\psi_1^*(u(s))] + \epsilon_2 \mathbb{E}_{\eta'}[\psi_2^*(v(a))]$$

$$= \min_{u,v,Q} \left\{ -\min_\mu \langle-\mu, r - A^{\pi*}Q - \epsilon_1 u\mathbf{1}_A^T - \epsilon_2 \mathbf{1}_S v^T\rangle + D_\psi(\mu|\mu') \right\} + \mathbb{E}_{b^\pi}[Q(s,a)] + \epsilon_1 \mathbb{E}_{\rho'}[\psi_1^*(u(s))] + \epsilon_2 \mathbb{E}_{\eta'}[\psi_2^*(v(a))]$$

$$= \min_{u,v,Q} \mathbb{E}_{\mu'}\left[ \psi^*\left(r(s,a) - A^{\pi*}Q(s,a) - \epsilon_1 u(s) - \epsilon_2 v(a)\right) \right] + \mathbb{E}_{b^\pi}[Q(s,a)] + \epsilon_1 \mathbb{E}_{\rho'}[\psi_1^*(u(s))] + \epsilon_2 \mathbb{E}_{\eta'}[\psi_2^*(v(a))], \tag{40}$$

and wrapping equation 40 with $\max_\pi$ results in equation 19. The gradient in equations equation 21, equation 22 and equation 23 are basic calculus. Assuming $Q^*, u^*, v^*$ are optimal functions out of the policy evaluation in equation 40 for the given fixed $\pi$, then using Danskin's theorem Bertsekas (1999) we have $\partial_\pi \min_{u,v,Q} \mathcal{L}(u, v, Q; \pi) = \partial_\pi \mathcal{L}(u^*, v^*, Q^*; \pi)$ and gradient in equation 24 is derived using the facts: **(A)** $P^\pi Q(s, a) = \mathbb{E}_{s' \sim P(\cdot|s,a), a' \sim \pi(\cdot|s')}[Q(s', a')]$ and **(B)** for any distribution $z \sim p$, $\partial_p \mathbb{E}_p[h(z)] = \mathbb{E}_p[h(z)\nabla \log p(z)]$.

In order to derive the objective of equation 32, first we interchange $\min_\mu$ and $\max_{u,v,Q}$ in equation 39 by minimax theorem. Fixing policy $\pi$, we rewrite the first term in equation 39 as

$$\langle \mu, r - A^{\pi*}Q - \epsilon_1 u \mathbf{1}_{\boldsymbol{A}}^T - \epsilon_2 \mathbf{1}_{\boldsymbol{S}} v^T \rangle = \mathbb{E}_\mu \left[ r(s, a) - A^{\pi*}Q(s, a) - \epsilon_1 u(s) - \epsilon_2 v(a) \right]$$
$$= \mathbb{E}_{\mu'} \left[ \zeta(s, a) \left( r(s, a) - A^{\pi*}Q(s, a) - \epsilon_1 u(s) - \epsilon_2 v(a) \right) \right],$$

where $\zeta(s, a) = \frac{\mu(s,a)}{\mu'(s,a)}$.

Equations equation 34–equation 37 are basic calculus derivations. For equation 38, given optimized $Q, u, v$ and $\zeta$, using **(A)** and **(B)**, we can take the gradient of the first two terms in equation 39, i.e.,

$$\mathbb{E}_{\mu'} \left[ \zeta(s, a) \left( r(s, a) - A^{\pi*}Q(s, a) - \epsilon_1 u(s) - \epsilon_2 v(a) \right) \right] + \mathbb{E}_{b^\pi}[Q(s, a)],$$

with respect to $\pi$ and combine them under the relation $\mu(s, a) = (1 - \gamma)p_0(s)\pi(a|s) + \gamma\pi(a|s)\sum_{s',a'} P(s|s', a')\mu(s', a')$ to get equation 38.

These two approaches in Section 4, are similar to policy gradient derivations in Nachum & Dai (2020), with corrective terms on $u(s)$ and $v(a)$ in the softmax operator defined in the main text.

# E   Demonstrations

## E.1   GridWorld Setup

In Section 5, we set $D_{\psi_1} = D_{\psi_2} = \text{KL}$ and we applied Dykstra in the gridworld. We used the Frobenius norm on the difference of two consecutive matrices out of Dykstra until the error is less than $10^{-5}$.

In order to see the extreme effect of $\rho'$ independently (setting $\epsilon_2 = 0$), we can enforce it as $\delta_{\rho'}(\mu 1)$, even though it might not be possible to find a $\mu$ such $\mu\mathbf{1} = \rho'$ as discussed in Section 5. In this setting, we observed Dykstra gets stuck switching back and forth between projection onto $\rho'$ and projection onto occupancy measures $\Delta$ and we get division by zero exception. In our experiments we observed similar outcome when using the penalty function $\text{KL}(\mu\mathbf{1}|\rho')$ with high coefficient $\epsilon_1$ (close to 20).

## E.2   Cart-Pole setup

For this test we set $\gamma = .99$, with $\beta_1 = 0.005$ and $\beta_2 = 0.001$, to compute the expectation with respect to policy, we averaged $Q$ values over 10 randomly sampled actions drawn from the policy at each state. To better see the effect of $\eta'$, we set a high value of $\epsilon_2 = 1000$. Policy in this example is a gaussian MLP network of two hidden layers of 8 units and $Q$ and $v$ have two hidden layers of 32 units.

