# OpenReview forum: "Policy Optimization with Distributional Constraints: An Optimal Transport View"
_TMLR — Rejected by TMLR_

### Review · Reviewer_kWXk · 2022-08-03

**Summary Of Contributions:**

The paper proposes a unifying objective for RL featuring constraints on state visitations and global action executions. The authors show that this new objective can be casted as a special case of unbalanced optimal transport. The canonical Dykstra's algorithm can then take over the optimization, provided that underlying dynamics of the MDP is known. In case we have no knowledge but samples from the MDP, the authors propose an actor-critic style algorithm.

**Requested Changes:**

See weakness above

**Strengths And Weaknesses:**

Strengths:
* The new regularized objective with constraints state and action visitation appears novel to me
* The use of Dykstra's algorithm to solve the new objective appears novel to me

Weakness:
* For the use of Dykstra's algorithm in Eqs 15 - 17, the authors acknowledge that Eq 17 does not have a closed form solution but does not clarify how to solve it. If the authors do not explicitly clarify how the projection in Eq 17 can be solved, I do not think this is a successful application of Dykstra's algorithm. I expect to see a computationally affordable method for solving this projection with convergence and performance guarantee. If it cannot be solved exactly, I expect to see how the errors resulting from the solver affect the overall convergence of the Dykstra's algorithm. Otherwise, I do not see the necessity for introducing Dykstra's algorithm.
* The derivation of the actor critic algorithm in Section 4.1 seems straightforward to me and follows easily from Nachum & Dai (2020) but I do not think we actually have access to unbiased samples for the gradient in Eq 21. In Eq 21, we do not know h since it involves the transition kernel P_\pi. But in the gradient in Eq 21, we need the weighted softmax of h. Could the author explicitly clarify how to obtain samples for this gradient? One solution I can think of is to introduce two-timescale learning and use an additional function approximatior to learn h.
* The motivation of introducing off-policy learning is not clear to me. IMHO the paper's contribution is about state and action visitation constraint. On-policy setting seems to be enough for me to demonstrate this contribution. So IMHO the authors should do an on-policy actor critic from Eq 18. Introducing off-policy learning in Eq 11 seems to make the demonstration of the contribution unnecessarily harder. I expect to see some clarification regarding this.
* I think the authors should avoid the use of the word large-scale across the paper. I feel in most RL communities, cart-pole is not considered large scale.
* Some notations should be clearly defined. For example, in the first equation of Section 2.1, I can guess \mu 1_Y is the marginal distribution of \mu on X (correct me if I am wrong). But I can't find a clear definition of it. Maybe this is something common in the authors' community. But this is not so common in the RL community, AFAIK

---

> ### Author Response · Authors · 2022-08-31
> **Answer to questions and comments**
>
>
> Thank you  for your thoughtful review and finding novelty in our work, we respond to your comments and questions as follows:
>
> - For the use of Dykstra's algorithm in Eqs 15 - 17, the authors acknowledge that Eq 17 does not have a closed form solution but does not clarify how to solve it. If the authors do not explicitly clarify how the projection in Eq 17 can be solved,....
>
> As explained in the paper, we think the main goal of viewing RL as OT was to come up with objective~7 and a systematic approach (Dykstra's) as a method of iterative projections to optimize it.
> Interestingly, Dykstra's and the large-scale algorithm in section 4 are technically the dual form of each other. That is, given a fixed policy, the policy
> evaluation problem is the dual form of the Dykstra's. So it is interesting to see that Dykstra's algorithm which is generalization of famous  Sinkhorn-Knopp algorithm in OT has this dual minimax policy evaluation formulation in RL that is explained in the paper and is also proposed in modern algorithms such as  DualDICE and algeadice. We have added a paragraph after equation 21 to the paper to address this concern of yours.
>
> - The derivation of the actor critic algorithm in Section 4.1 seems straightforward to me and follows easily from Nachum & Dai (2020) but I do not think we actually have access to unbiased samples ....
>
> In offline setting with samples from distribution $\mu'$, in eq 21(Now eq 24), the transition kernel doesn't need to be known as we can approximate $P^\pi Q(s,a)$ with samples of $Q(s',a')$ where $s'$ is the next state sample after executing $(s,a)$ and  $a' \sim \pi(\cdot|s')$. So the $P^\pi$ wont cause any problem, since it samples from $\mu^\pi$. Now, we have an outer expectation wrt $\mu'$ and the same in the denominator of $F \circ h$. assuming samples of $\mu'$ are in batch we shouldn't have problems estimating these expectations in an unbiased way and form the division.
>
>
> - The motivation of introducing off-policy learning is not clear to me. IMHO the paper's contribution is about state and action visitation constraint....
>
> Thank you for your honest opinion and we think you are right that the offpolicy scenario might disturb the flow of the paper and for this reason and your suggestion  we keep it in the appendix.However, it is needed in case when the when $\rho'$ and $\eta'$ are derived from $\mu'$, that is the constrint distributions are the marginals of $\mu'$. in this case because we need to take advantage of fenchel duality if we directly add $KL(\mu|\mu')$ to the objective, it would make our marginal constraints redundant because minimizing $KL(\mu|\mu')$ already means minimizing $KL(\eta|\eta')$ and $KL(\rho|\rho')$. So we use the importance weights to transform the $E_\mu$ to $E_{\mu'}$(because we have samples from $\mu'$) and take the extra effort to approximate those importance weights.
>
> - I think the authors should avoid the use of the word large-scale across the paper. I feel in most RL communities, cart-pole is not considered large scale. Some notations should be clearly defined. For example, in the first equation of Section 2.1, I can guess$ \mu 1_Y$ is the marginal distribution of $\mu$ on X (correct me if I am wrong). But I can't find a clear definition of it. Maybe this is something common in the authors' community. But this is not so common in the RL community, AFAIK
>
> Cart-pole is definitely not large-scale in Rl community, and this is why we called the algorithm large-scale and we called our tests "demonstrations", because our goal in the paper was to show this connection of RL and OT can result in the marginal constraints and granular exploration of these constraints was more important.
> In terms of notation, in notation section we defined that $1_Y$ is the vector of all ones on space Y, so as you mentioned correctly $\mu_{\mathcal S\times \mathcal A}1_A$ is the marginal on $S$.

---

### Review · Reviewer_Ue3n · 2022-08-15

**Summary Of Contributions:**

This paper presents an algorithm for policy optimization in a setting where the user desires to implement constraints on the marginal distributions of either the state visitation distribution or the global action distribution (i.e. averaged over visited states). The algorithm itself is inspired by optimal transport and first presented in tabular environments with known dynamics and then relaxed to larger environments with an actor-critic version. Some toy experiments show that changing the constraints indeed changes the learned policies.

**Requested Changes:**

Substantive changes:
1. (Necessary) Add a better description of the motivation for this particular version of constraints in the introduction.

2. (Necessary) Add a more detailed comparison to existing methods and discussion of why existing methods and naive baselines do not suffice in the setting of RL with distributional constraints on the marginals.

3. (Necessary) Add more clear takeaways from the experiments and more quantitative descriptions of the effect of the proposed algorithm relative to some baseline approaches.

4. (Not absolutely necessary, but would go a long way to improve the paper) Either add (a) larger scale experiments with proper baselines or (b) theory demonstrating convergence rates and sample complexity.

5. (Not absolutely necessary) The paper as written now spends much more time explaining the connections between OT and RL than actually presenting the novel contribution. It would likely make the contribution more clear if more of the paper focused on the key ideas of the contribution rather than the high level connection to OT. Perhaps much of the information is necessary to explain the contribution, but it seemed to me that much of it could be perhaps briefly explained in the main text and largely moved into the appendix since it is not the main contribution of the paper.

Typos:
- In the abstract "marginals are available" should be "marginals when they are available"
- After equation (2) "Entropy-regularized" should be "The entropy-regularized"
- In the second paragraph of section 6 "an constrained should be "a constrained"

**Strengths And Weaknesses:**

Strengths:
1. The paper proposes a clean and simple algorithm for the proposed problem setting of RL with cnstraints on the marginal distributions over states and actions. The algorithm is a nice extension of ideas from OT into RL in a natural way.

2. The derivations of the algorithm all seem to be sound and everything seems to be well-defined even if the notation is sometimes a bit heavy.

Weaknesses:
1. The paper lacks motivation for the problem setting. The main contribution is to provide an algorithm for constrained RL where the constraints take a particular form (in terms of constraints over the marginals over either states or actions). As written now, it seems that the paper starts from the idea to connect RL to OT and then comes up with whatever style of constraints would be convenient for the OT formulation rather than actually making a clear argument that these kinds of constraints are natural or necessary in real problems. Without a clearer motivation it is difficult to understand why the algorithm is interesting. Some concrete examples of where this type of constraint is useful could be helpful.

2. The paper lacks convincing arguments that novel algorithm is needed for this setting or outperforms existing algorithms if they were adapted to this setting. I understand that the proposed algorithm is a valid solution to the proposed problem, but am not quite sure that other methods would not work here. This is somewhat discussed in the related work section when considering constrained MDPs, but the arguments is not entirely convincing that similar performance could be achieved using existing ideas from that literature. Especially in the experiments there are no baseline methods presented whatsoever. It would even be useful to see why simple naive baselines like adding penalties to the reward function proportional to constrain violation and then just running standard RL are not able to achieve similar results to the proposed method.

3. The experiments are quite small in scale, primarily qualitative in nature, and do not have any baselines. It is ok to just have small scale experiments if they are very illustrative, but these ones are not, so they should either be improved in terms of scale or clarity. In terms of clarity, it would be nice to see quantitative comparisons to some baseline approaches in terms of some relevant performance metric. The current approach of just illustrating pictures for different levels of constraint does not serve to convince the reader that this is a useful algorithm to apply in a new problem. Also, the description of imitation learning in cartpole as "a large RL task" in Section 5 is just false and this description should be removed.

4. There is a lack of more quantitative theoretical analysis of things such as convergence rates or sample complexity. The algorithm is primarily presented in the tabular setting with known dynamics (and thus is more strictly considering the planning/MDP optimization problem rather than standard RL where the dynamics are unknown) so convergence rates would be desirable. Then the actor-crtic algorithm is presented as a heuristic that can be extended to unknown dynamics by replacing expectations with samples. Analyzing the sample complexity of this algorithm would also be interesting.


Overall summary:
I think this is a technically sound paper, but is lacking in terms of motivation which can hopefully be improved in a revision as detailed in the requested changes. Moreover, to take this from a marginal paper to a strong paper, the addition of either larger scale experiments or tighter theory would go a long way.

---

> ### Author Response · Authors · 2022-08-31
> **Answer to questions and comments**
>
> Thank you for your detailed reviews and thoughtful comments. We answer to your requested changes as they share answers with your comments.
>
>
> - Add a better description of the motivation for this particular version of constraints in the introduction.
>
> In terms of motivation of these new constraints and flow of ideas in the paper, as you mentioned the goal is to connect OT and RL, or in better words -- see RL from the viewpoint of OT. To do so, we proposed the objective 7 for RL in the paper which is a general form of the U(OT) problem. This objective is general enough to instantiate the 3 main objectives in  policy optimization literature as well by varying $\Gamma$ and $\phi_i$'s. Besides, the OT view motivates new type of distributional constraints in RL that has three applications in RL: 1- Regularizers in iterative policy optimization scenarios which we extensively discussed after Eq~13; 2- problem specific constraints on actions budget ans safety constraints on states (Please note that they don't necessarily need to be distributions and as we discussed any positive measure can be used for these types of constraints) and 3- imitation learning (e.g., the cart-pole demonstration ).
>
> Also, to address your concern on motivation of these types of constraints,we added the following sentence in the introduction:
> Marginal constraints of $\rho'$ and $\eta'$ can arise in various settings in policy optimization: problem constraints of budget on actions or safety constraints on state visitations, empirical state and action distributions from an expert demonstration or for regularization purposes in an iterative policy optimization algorithm.
>
>
>
> - Add a more detailed comparison to existing methods and discussion of why existing methods and naive baselines do not suffice in the setting of RL with distributional constraints on the marginals.
>
> This is a great question and we have addressed the inefficiencies/differences of existing constrained policy optimization in the "Objectives and Constraints in Reinforcement Learning" part of Section 6.  The major difference of conventional constrained policy optimization and the type of distributional constraints that we are proposing  is that we are considering constraints in a more finer level, that is the traditional cost constraints in RL is summarized  in  $E_\mu(c)\leq c$ for some constant $c$ and cost function $C(s,a)$. However, the distributional constraints are capable of bounding the marginals over states and actions. These types of constraints can be flexible as they just need to be positive measures and not necessarily distributions and we can have hard constrains (instead of KL) as discussed in "other settings" of section 6.
>
> - Add more clear takeaways from the experiments and more quantitative descriptions of the effect of the proposed algorithm relative to some baseline approaches.
>
> Thanks for your comment we added some description of the objective 20 and corresponding algorithm with constraints(about cost functions $u,v$ , the terms in the objective and adjusted reward function and temporal difference error) in comparison to non-constrained policy optimization  after objective 20 in sec 4 that we think addresses your concern.
>
> -(Not absolutely necessary, but would go a long way to improve the paper)Either add (a) larger scale experiments with proper baselines or (b) theory demonstrating convergence rates and sample complexity.
>
> Thanks for suggesting the idea, however, considering the limited time to respond to reviews we are unable to provide such results as they require more time on detailed analysis of convergence rate  and sample complexity. However, in case of more large scale experimentation, we are not convinced how this would be helpful as we are not suggesting an algorithm that outperforms existing ones in a specified measure. This is why we experimented on two simple tasks at a very granular level to show how Dykstra's and the corresponding sample-based algorithm can be helpful and the distribution constraints can produce meaningfully results.
>
> -(Not absolutely necessary) The paper as written now spends much more time explaining the connections between OT and RL than actually presenting the novel contribution. It would likely make the contribution more clear if more of the paper focused on the key ideas of the contribution rather than the high level connection to OT. Perhaps much of the information is necessary to explain the contribution, but it seemed to me that much of it could be perhaps briefly explained in the main text and largely moved into the appendix since it is not the main contribution of the paper.
>
> Thanks for your suggestion, having the goal of view RL as OT, we  tried to move the technical parts to the appendix as much as we can to help the flow of the paper, we are open to any specific suggestions you have to improve the readability of the paper.

---

### Review · Reviewer_bNWT · 2022-08-17

**Summary Of Contributions:**

This paper considers to formulate policy optimization as a constrain optimization problem, where the constraints are on the marginal state stationary distributions and action distributions. Based on the new formulation, the authors can view policy optimization as an unbalanced optimal transport problem, and proposed to solve it using Dykstra's algorithm. Besides, the authors also discuss how the proposed method can be adapted to an actor-critic algorithm when the state or action space is large.

As a summary, the main contributions of this paper are the follows: 1) proposing a general purpose RL objective based on Bregman divergence 2) proposing to use Dykstra's algorithm to solve the problem. 3) providing demonstrations to show the effectiveness of the proposed approach.



**Broader Impact Concerns:**

N/A.

**Requested Changes:**

1. Writing and discussion on different Bregman divergence. I wish authors could add more dicussion on the choice of Bregman divergence in your framework. For both methodology discussion and empirical experiment comparison. This is quite important. (necessary)

2. I wish there are more empirical experiments, especially continuous experiments on mujoco to verify your algorithm's performance and stability. Moreoever, I hope there are some experiments comparing related algorithms, such as algaedice. If you think there is no need to compare, please clarify why. (necessary)

3.  The authors only use KL divergence to do the regularization. So under this case, what is the difference between your algorithm (using Dykstra’s algorithm) to solve the optimization problem and directly using mirror descent to solve the problem. I would appreciate if the authors could have a discussion. (necessary)

4.  Abalation study on choice of hyperparameters such as $\epsilon_1$ and $\epsilon_2$. (necessary)

5. Can you have detailed algorithms box to describe your algorithm? for both the discrete case and continous case. Please don't refer too many equations in your paper in the algorithm box. It is quite difficult for readers to understand. For the continous case, you may need to write down the object for each variables, which could be helpful for others to understand. (necessary)

6. I believe the additional regularization term introduce benefits theoretically. However, it also introduce additional optimization difficulties.  Can you have a comment on the tradeoff between the algorithm complexity and optimization difficulty? Still, I think the optimization difficulty may degenerate your algorithm empirical performance. (necessary)


7. Please use citep to fix your citation problem. (not necessary)

8. $D_{\psi_1}$ and $D_{\psi_2}$ in Eq(13) are opposite in your above paragraph, compared with those in Eq(123). (not necessary)

**Strengths And Weaknesses:**

Strengths: The new objective is kind interesting, which can be potentially motivated for deeper understanding of the connection between optimal transport and RL.  What's more, The authors also discussed how the new objective can generalize previous work, where we choose different instances of functions in Eqn (7). Also, the authors demonstrate how the proposed penalizations could be helpful and the empirical convergence performance when we use Dykstra's empirical performance.

Weakness
- The objective Eq.(7)  is interesting. However, I believe the authors only discuss when $\Gamma$, $D_{\psi_1}$, $D_{\psi_2}$ are KL. There is not much discussion on choosing different $D_{\Gamma}$ and $\phi_i$. In that case, the proposed general objective seems to lack justification and I can not see why you want to start with the general objective. Also, If you only discuss the case of KL divergence in your algorithm, the writing can be simplified and readers can understand your algorithm in a more easier way, since the additional formulation does not introduce much benefits for readers to understand.
- When the state and action space are large, and we can not directly obtain $\pi$ from $\mu$, the authors propose to optimize policy pi using a max-min optimization over four variables, which are very complex and I doubt it could be applied to relatively complex cases such as mujoco environment,
- The new algorithm introduces more hyperparameters, such as $\epsilon_1$ and $\epsilon_2$. I concerns that it may require additional hyperparameter tunning for each new environment, which could bring additional difficulty for the practical policy optimization, especially when you already introduce complex minimax optimization here.
- The experiments are relatively weak. most of the experiments are conducted to verify the correctness and effectiveness of the algorithm. For example, examining whether the proposed regularization terms are effect or not.
- I think your current proposed algorithm is quite related to prior works such as dual-critic[1], sbeed[2] or algaedice[3]. There are very few discussion in the main content (except few discussion in the related work).


---
[1]  Dai, Bo, et al. "Boosting the actor with dual critic." arXiv preprint arXiv:1712.10282 (2017).

[2] Dai, Bo, et al. "SBEED: Convergent reinforcement learning with nonlinear function approximation." International Conference on
Machine Learning. PMLR, 2018.

[3] Nachum, Ofir, et al. "Algaedice: Policy gradient from arbitrary experience." arXiv preprint arXiv:1912.02074 (2019).

---

> ### Author Response · Authors · 2022-08-31
> **Answers to comments and questions**
>
> Thanks for finding our paper interesting and you thoughtful comments.Following we answer your comments and questions:
>
> -  The objective Eq.(7) is interesting. However,..
>
> The goal is to connect the two theories. As we discuss after eq 8, we categorized policy optimization objectives with different $\phi$'s and $\Gamma$'s that results in objectives 3,5,6. There introduce two choices of $\Gamma$ (entropy and normalized entropy), which result in two different $D_{\Gamma}$ in objectives 3 and 5. Similarly for the choices of $\phi's$ we have introduced various $\phi$'s that result in objective 3,5,6 and also we introduced the distributional constraints as new $\phi$'s. One choice for $\phi$ is $D_\psi$ which as you referenced results in AlgaeDICE, DualDICE algorithms (e.g., with $\psi = x^2$).  It is an interesting direction for the paper to find new $\Gamma$'s and $\phi$'s to come up with new RL objectives.
>
> Please see the answer to the rest of your comments in the requested changes as they share answers.
>
> Requested Changes:
>
> 1-Writing and discussion on different Bregman divergence...
>
> In pager 4 after eq 8, we have categorized policy optimization objectives with different $\phi$'s and $\Gamma$'s that results in objectives 3,5,6. There we  introduced two choices of $\Gamma$ (entropy and normalized entropy), which result in two different $D_{\Gamma}$ in the literature that result in objectives 3 and 5. Choice of $\Gamma$ is a tricky one as we'd like to use the fenechel duality in a way that we get an expectation form out of it. So far we have categorized them in these two choices of $\Gamma$ that are evaluated empirically and in depth in Neu et.al.(2017), we agree with you that it would be an interesting direction of this work to see what other choices of $\Gamma$ could be possible in deriving a new objective.
>
> 2-I wish there are more empirical experiments...
>
> We think comparing with baseline helps the RL community if we were trying to show that our algorithm is improving some performance measure (e.g., total return), however, our goal was to connect the two theories first and show the general objective 7 can categorize and even result in new RL objectives and various type of constraints like distributional case which is motivated by OT. In empirical evaluation we focused on the constraints rather than the variation of unconstrained RL objective  such as algaedice and work of (Nachum \& Dai 2020). it would be interesting to combine algaedice with these distributional constraints but it was not the focus of current paper.
>
> 3-The authors only use KL ..
>
> In case of regularization, assuming we use KL for both case, distributional constraints result in different approach than mirror descent. We discuss in section 4, when $D_{\psi_1}=D_{\psi_2}=$KL, objective 13 looks similar to 3 (considering expansion in 4), but they are different. In 13, if $\rho' = \mu'{1}$ and $\eta' =\mu'^T{1}$, for some baseline $\mu' \in \Delta$, then the third term is KL $\left (E_{\rho^\pi}[\pi]\mid E_{\rho^{\pi'}}[\pi']\right)$ which is a global constraint on center of mass of $\pi$ over the whole state space, whereas $E_{\rho^{\pi}}[\text{KL}(\pi\mid \pi')]$ in 5 is a stronger constraint on closeness of policies on every state. The bottom line is that  we are imposing a relaxed condition on row-sum and column-sum of $\mu'$.
>
> 4-Abalation study  on $\epsilon_1$ and  $\epsilon_2$.
>
> You are right, $\epsilon_1,\epsilon_2$ could bring more complexity. However, one could remedy this by setting $\epsilon_1=\epsilon_2$ for hyperparameter tuning. This is an interesting part of the algorithm and allows you to put different emphasis on action distribution and state distribution depending on the nature of your problem. for example in our cartpole problem we set the $\epsilon_2=0$ . we have also done this ablation study for the little grid world example already at a granular level.
>
> 5-Can you have detailed algorithms box ...
>
> We have modified section 4 to address your issues, we also added the interpretation of the objective functions for a better reach.
>
> 6-I believe the additional regularization term introduce benefits theoretically..
>
> Thanks for such a smart question.You are right, in case of regularization when $\rho'$ and $\eta'$ are from $\mu'$ ($\mu'=\mu_{k-1}$ in iteration $k$) complexity of our proposed method is higher as we need to approximate $\zeta,u,v$. one can reduce the complexity by only regularizing at row-sum or column sum and have two approximators rather than three. However, we believe it is the trade off for having a more relaxed regularizer on the RL objective. it is definitely an interesting path to explore. Also, it is important to mention that in case of imitation learning and distributional costs which is the algorithm in the main text as we don't need to approximate $\zeta$ and assuming constraining on row or column sum of $\mu$, we only need one extra function approximator other than $Q$.

---

### Review · Reviewer_GvBr · 2022-08-17

**Summary Of Contributions:**

The paper makes a certain connections between policy optimization objective in RL and unbalanced optimal transport. Concretely, the paper considers the policy optimization formulation with occupancy measures as the primal optimization variable, and instantiates equality constraints in unbalanced optimal transport as the recursive constraints for the occupancy measures. Such constraints can also include entropy constraints or trust region constraints. The paper proposes to use Dykstra's algorithm for solving the optimization problem in the tabular setting, and proposes a sample based algorithm for deep RL. The paper tests out the algorithm on tabular domains and cartpole domain, showing sensible behavior of the proposed algorithm.

**Broader Impact Concerns:**

No impact concerns.

**Requested Changes:**

#### **Presentation**
I hope the authors can improve the presentation of the work, by offering proper explanations to complicated notations such as those in Eqn 21. See comments above about weaknesses of the work.

#### **Algorithm**
It is better to describe algorithm 1 in the main paper, as it should constitute a core algorithmic contribution. However, as stated above, the sample-based algorithm 1 in Sec 4.2 is not novel enough as it can be derived from the objective using saddle point optimization technique. The perspective of OT is not clear, and it is not clear why adopting the OT view is interesting nor useful.

#### **Dykstra's**
A flagship result here I expect is to adapt Dykstra's algorithm to sample-based RL setting. Unfortunately, the current paper does not show such adaptation and only argues that Dykstra's algorithm is challenging to apply. In general, I find this to be quite a shame because the central novelty of this work would be to try to formulate RL as OT problem and adapt an OT algorithm for the RL case. The paper should strive to demonstrate this.

#### **Experiments**
Comparing to cartpole is not enough. Sec 4.2 talks about large-scale RL but there is nothing significant about cartpole experiment that demonstrates the large-scale nature of the domain. To showcase the efficiency of the proposed method, the paper should consider additional more high-d domains.

Further the paper should compare with reasonable baseline and prior work. A simple baseline would be to run RL without the constraint loss and see that the algorithm does violate the constraints, which the current algorithm can avoid.

**Strengths And Weaknesses:**

The paper's strengths lie in identifying the connection between RL objective and unbalanced optimal transport. From a theoretical point of view, such connections are interesting in themselves; algorithmically, such connections entail that algorithms for solving optimal transport be adapted to the RL context, potentially leading to more efficient learning algorithms.

In my view, the paper's weaknesses lie in a few aspects, which I detail below.

#### **Presentation and notation can use improvement**
The paper can potentially make more efforts in adjusting optimal transport literature's notation to the RL context. In its current form, I would feel that it is in general a bit difficult for RL audience to parse the notations. Certain notations are also just plainly spelled out in the main paper, without enough explanation and elaboration, such as Eqn 21.

#### **Insufficient novelty of the algorithmic contribution**
Relating RL to unbalanced optimal transport is interesting, as it entails the application of Dykstra's algorithm to the RL context. However, a core feature of RL is sampled based learning, which the Dykstra's algorithm does not address out of the box. Indeed, most materials up until Sec 4.1 do not introduce much element of sample-based learning. In Sec 4.2, the paper proposes a primal-dual saddle point optimization formulation to optimize the objective, bearing little connection to Dykstra's algorithm. Indeed, the authors themselves argue in Sec 4.2 that Dykstra's algorithm is challenging to be applied to sample-based learning setting. The saddle point optimization algorithm formulation is quite straightforward given prior work such as Nachum et al, and Dai and Nachum, as the authors noted in the prior work, though there are some technical differences such as that this work offers an OT interpretation.

In summary, the algorithmic contributions in the paper as they are now, are not novel nor significant. Dykstra's algorithm cannot be applied to sample-based learning, and cannot be adapted accordingly. The algorithm in Sec 4.2 is detached from Dykstra's algorithm and is not uniquely defined by the OT problem itself.

#### **Experiment not enough**
Most experiment efforts are on the tabular domain, where the MDP information is accessible to the Dykstra's algorithm. On top of that, in Sec 5.2, only cartpole is tested. Cartpole is one of the simplest RL domain, and cannot even be interpreted as proper deep RL benchmark (for which higher-d problems and more challenging domains are warranted). There is also no comparison to related work nor baseline.

---

> ### Author Response · Authors · 2022-08-31
> **Responding to the comments and questions**
>
> Thanks for your detailed comments and finding our paper and connection of OT and RL interesting. We have answered your concerns and comments as follows:
>
> 1- Presentation and notation can use improvement:
>
> Please see the response in part one of the request changes.
>
> 2-Insufficient novelty of the algorithmic
> Please see the response to part 2 of the requested changes
>
> Experiment not enough
>
> Please see the response to part 3 of the requested changes
>
>
> Requested Changes:
>
> 1-Presentation:
>
> We have modified the section 4 based on your requested changes. Now, Equation 21 (now Equation 24) and Objective 21 have more descriptions with regards to differences with regular policy optimization and the new constraints we consider in this paper.
>
> 2-Algorithm:
>
> Thanks for your suggestion, We brought the algorithms to the main text in section 4. With regards to the second part of your question on why OT view is adopted and why it is useful, in order to connect the two theories we needed to abstract away from both RL and OT. Optimal transport with hard constraint is solved through an algorithm that is called Sinkhorn-Knopp and if one wants to generalize the OT objective to unbalanced case or more than two set of constraints, Dykstra's algorithm as a method of iterative projections come into play and the RL Objective~7 that we proposed is the most general way that Dykstra's can solve. The usefulness of this objective is that in also include all three RL objectives in Eq 3,Eq 5,and Eq 6. So, The OT view has already helped to unify policy optimization objectives and a general objective that can be instantiated by varying $\Gamma$ and $\phi_i$'s. One special case this paper proposed is the distributional constraints, we hope RL community instantiate other objectives from it.
>
> In the matter of  usefulness of Dykstra's, it is interesting to mention that the  Dykstra's and the sample based algorithm in section 4 are the primal and dual form of the same optimization problem under fenechel duality- as verified by the proof of  Dykstra's in the appendix. So another novelty of this work other than connecting OT and RL at the objective level is to show that the Dykstra (e.g., Sinkhorn knopp) algorithm that is used in OT is in it's dual form similar to the policy evaluation problem corresponding to the saddle point optimization in RL that is proposed in this work and work of Nachum \& Dai(2020). To address this concern we explained  this connection after equation 20.
>
> 3-Experiments
>
> Comparing with baseline helps if we were trying to show that our algorithm is improving some performance measure (e.g., total return), however, as you mentioned if we already know that unconstrained RL would violate the the constraints, then what is the point of comparison? But if we study the effect of constraints on the policy, isn't it comparing with the unconstrained case and what we have already shown? for example in the cartpole example we know that we will not get such a bi-modal distribution because the expert policy distribution out of the policy gradient  is a Gaussian near zero. That is why we focused on granular explanation of the new distributional constraints in simple tasks that we thought could help demonstrate their applicability.  Also, our goal was not to beat any other algorithm in any performance measure, instead we tried to show how RL can be seen as an (U)OT problem, connection of Dykstra's and policy evaluation in saddle point optimization in RL, and showing the objective 7 is product of this view which can results in new objectives in RL which one of them is using the marginals on states and actions instead of considering the occupancy measures(joint distribution) as regularization or imitation learning constraint.

---

### Decision · Action_Editors · 2022-10-05

**Recommendation:** Reject

**Comment:**


This paper reveals a connection by reformulating reinforcement learning to optimal transport (OT), in which the state-occupancy constraints can be easily added. The resulted OT problem can be solved by tailoring Dykstra's algorithm in tabular settings.

All the reviewers acknowledge such an connection, however, there are still several common issues raised by the reviewers

1, The proposed connection between OT to RL did not bring justified benefits from either theoretical or empirical perspective (Reviewer bNWT).

2, More discussion to related work are needed to distinguish the proposed method (Reviewer GvBr) for a better position among the literature.

3, The proposed method is still difficult to generalize for realistic settings, e.g., double sampling (Reviewer kWxk, Ue3n), and therefore, no practical empirical comparison has been conducted (Reviewer bNWT,  Ue3n and GvBr).

In sum, these issues diminish the interests of the community on this paper.  After the rebuttal, all the reviewers still believe the current version of the paper are not ready for being published and reach the reject decision. I encourage the authors to consider the suggestions raised by reviewers to improve the paper for a better version.

**Audience:**


Yes, the paper can be potentially interesting to RL community, and potential OT community. But the current version did not provide convincing evidence to show the benefits of the connection.


**Claims And Evidence:**


The paper reveals the connection between between RL and optimal transport, which can be solved by Dykstra's algorithm.

However, the derived algorithm is not ready to be applied in practical settings. There is no convincing benefits (theoretical or empirical) yet from this new view. These drawbacks diminish the potential interests from the community.